# Functional impact of splicing variants in the elaboration of complex traits in cattle

Mathieu Charles [1,2,4], Nicolas Gaiani [1,4], Marie-Pierre Sanchez[1,4], Mekki Boussaha[1], Chris Hozé [1,3], Didier Boichard [1], Dominique Rocha[1] & Arnaud Boulling [1] ✉

GWAS conducted directly on imputed whole genome sequence have led to the identification of numerous genetic variants associated with agronomic traits in cattle. However, such variants are often simply markers in linkage disequilibrium with the actual causal variants, which is a limiting factor for the development of accurate genomic predictions. It is possible to identify causal variants by integrating information on how variants impact gene expression into GWAS output. RNA splicing plays a major role in regulating gene expression. Thus, assessing the effect of variants on RNA splicing may explain their function. Here, we use a high-throughput strategy to functionally analyse putative splice-disrupting variants in the bovine genome. Using GWAS, massively parallel reporter assay and deep learning algorithms designed to predict splice-disrupting variants, we identify 38 splice-disrupting variants associated with complex traits in cattle, three of which could be classified as causal. Our results indicate that splice-disrupting variants are widely found in the quantitative trait loci related to these phenotypes. Using our combined approach, we also assess the validity of splicing predictors originally developed to analyse human variants in the context of the bovine genome.

For more than a decade, the emergence of high-throughput genomic technologies has led to detailed knowledge about the sequence and variability of the bovine genome. This has resulted in the production of genome-wide association studies (GWAS) conducted directly on imputed whole genome sequence (WGS) data for numerous agronomic traits in cattle[1,2]. GWAS pinpoint a common variant statistically associated with a trait of interest; however, they provide no evidence about its causality. One corollary of the long-range linkage disequilibrium (LD) that exists in bovine breeds is that GWAS has mainly resulted in the detection of multiple variants in high LD rather than a single, truly causal variant[3]. Variability in imputation accuracy may also result in a more significant association for a variant in LD with the causal variant rather than the variant itself. This weakness inherent to statistical approaches makes difficult the identification of causal variants in most situations. It represents a major concern in bovine genomics because the integration of causal variants into genomic

evaluation models could generate more accurate predictions and sustain these models across generations, especially for distantly related animals[4].

A deeper understanding of the genome function has been considered to be relevant to help the identification of the variants underlying the phenotypes of interest in livestock[5]. In this way, the Functional Annotation of Animal Genomes (FAANG)[6] and FarmGTEx[7] initiatives are devoted to the production of functional genomic data of domesticated animals. For example, the Cattle Genotype-Tissue Expression (cGTEx) atlas spans over 100 tissues/cell types among over 40 breeds and reports thousands of *cis*- and *trans*-genetic variants associated with gene expression and alternative splicing for 24 major tissues in cattle[7]. The compendium of this type of data is expected to facilitate the identification of functional variants and explain the genotype-to-phenotype link in domesticated animals. For instance, strategies that aimed at integrating such kind of biological data in

[1]Université Paris-Saclay, INRAE, AgroParisTech, GABI, 78350 Jouy-en-Josas, France. [2]INRAE, SIGENAE, 78350 Jouy-en-Josas, France. [3]ELIANCE, 75012 Paris, France. [4]These authors contributed equally: Mathieu Charles, Nicolas Gaiani, Marie-Pierre Sanchez. ✉e-mail: arnaud.boulling@inrae.fr

GWAS were successfully employed to rank the functional importance of genetic variants across the bovine genome[8,9]. Nevertheless, it is worth noting that making predictions from indicators of functions (e.g. gene expression, chromatin accessibility, sequence conservation) is limited in comparison with accurately assessing the functional impact of variants. A variant located in a functional region of the genome is more likely to have an impact on its function, leading to phenotypic consequences, but this information is not in itself proof of the functional nature of the variant. For example, a variant located in a regulatory region, such as a promoter, will be assigned a higher probability of being causal than variants located outside any functionally annotated regions. However, such a promoter variant may well have no effect on the expression of the gene under the control of this promoter, as illustrated by numerous studies (e.g. rs79134272, rs4765182[10] and rs10273639[11]).

Splicing is the process by which introns are removed from the primary messenger RNA (mRNA) transcript, and the exons are joined together to obtain a mature mRNA[12,13]. Alternative splicing refers to the process where different combinations of exons from the same gene can be joined or skipped, resulting in diverse mRNA transcripts that encode proteins with varied structures and functions. This mechanism allows for greater complexity of the proteome and participates in phenotypic diversity[12,14]. It also controls mRNA transcript abundance through the non-sense mediated mRNA decay (NMD) and other RNA degradation mechanisms[15,16]. Therefore, alternative splicing represents a central element in gene expression, and it often occurs in a developmental, tissue-specific or signal transduction-dependent manner[16,17]. Transcriptomic studies have shown that alternative splicing is prevalent across eukaryotes and, for instance, affects the expression of 90 to 95% of human genes[18,19]. In addition, genetic mutations are an important driver of altered gene expression and may generate novel splice patterns, thus contributing to the emergence of alternative mRNA transcripts[14,20].

Genetic alterations occurring in the DNA sequence of a gene and modifying the normal splice-processing of its precursor RNA are called splice-disrupting variants (SDV). This type of variant results in a modification of the mature RNA sequence by abnormal inclusion or exclusion of exonic or intronic regions from the precursor RNA. SDV have been widely studied for decades and are a major cause of Mendelian disorders[21,22]. The role of human SDV in the elaboration of complex phenotypes has remained more elusive until recent studies on the impact of RNA splicing on the modulation of phenotypic traits. Li et al. observed that genetic variants that are associated with variation in the splicing ratios of transcripts (sQTL) exhibited effects of similar or even larger magnitude than genetic variants that are associated with variation in the expression level of transcripts (eQTL)[23]. Recently, Ting et al. observed that eQTL and sQTL each explained a distinct fraction of the heritability of complex traits in humans, of about 10%[24]. A major effect of RNA splicing in shaping complex traits has also been reported in cattle. Xiang et al. showed that sQTL explained up to 66% of trait heritability, of which nearly 60% were directly related to *cis*-sQTL[25]. By contrast, eQTL explained 50% of trait heritability, of which only 30% were directly related to *cis*-eQTL. sQTL are not exclusively linked to SDV, as variants that alter the transcription rate can also have an effect on splicing mechanisms[26]. However, all these observations strongly suggest that SDV play an important role in the construction of bovine complex phenotypes. Apart from that, few studies have directly documented the effect of SDV in cattle. To our knowledge, 17 SDV responsible for monogenic diseases[27–43] and 7 SDV involved in complex traits[44–50] in cattle have been identified.

Large-scale functional validation of SDV represents a technical challenge that can be addressed since the advent of massively parallel reporter assays (MPRA). These technologies are suitable to discriminate functional variants from non-functional ones. After being initially used to assess *cis*-regulatory elements[51–53], MPRA have been successfully used to validate human SDV through different methods[54]. The MaPSy method was first developed but did not allow the analysis of intronic variants[55,56], by contrast with Vex-seq and MFASS methods[57,58]. Along with experimental assays, splicing prediction programs have been developed and mainly used to guide molecular diagnosis of human diseases[59]. In recent years, progress realised in the field of artificial intelligence has increased the performances of these tools by means of deep-learning-based methods, as illustrated by SpliceAI and Pangolin[60,61]. These two programs demonstrated high accuracy and outperformed their predecessors in predicting the spliceogenicity of genetic variants, which is their ability to impact the splicing of the gene in which they are located.

The objective of this study was to analyse a large number of bovine candidate SDV in order to characterise them and understand their impact on complex traits. After performing GWAS on various bovine phenotypes, we used a combination of high-throughput in silico (i.e. SpliceAI and Pangolin) and experimental (i.e. Vex-seq) tools to predict and validate SDV in the bovine genome. Some of them were identified as putative causal variants for these phenotypes. Moreover, we used the experimental data generated with the Vex-seq method to assess the performance of SpliceAI and Pangolin when used with the bovine genome.

## Results
### GWAS summary

GWAS were conducted on imputed WGS data to investigate 20 traits associated with milk production and composition, fertility, mastitis resistance, growth, as well as live and carcass morphology. The analyses included diverse populations of animals, comprising 2255 to 10,066 bulls, cows or steers from four distinct bovine breeds, one dairy breed (Holstein), one beef breed (Charolaise), and two dual-purpose breeds (Montbéliarde and Normande) (Fig. 1 and Table 1). To account for breed-specific effects, each GWAS analysis was performed separately for 48 breed x trait combinations.

From a comprehensive set of 25 million biallelic SNP that were tested, after removing variants with the lowest frequencies (MAF <0.005), variants with $-\log_{10}(p\ value) > 6$ were selected as potential causal variants. This selection process led to the identification of 218,723 trait × breed × variant combinations, representing a total of 138,971 unique variants, hereafter named candidate variants, distributed across the entire bovine autosomes (Supplementary Table 1). Notably, chromosome six harboured the largest number of variants (43,613), followed by chromosomes 14 (21,526), 2 (18,712) and 5 (14,241).

Across the Holstein, Montbéliarde, Normande and Charolaise breeds, a total of 10, 15, 15 and 8 different traits were analysed, respectively. The Holstein breed exhibited the highest number of candidate variants (57,551), followed by Charolaise (46,949), Montbéliarde (42,332) and Normande (36,512); these variants were located in 232, 138, 108 and 95 QTL regions, respectively. In the three dairy or dual-purpose breeds (Holstein, Montbéliarde and Normande breeds), the majority of candidate variants were associated with milk protein content (34,140, 25,334 and 19,751, respectively) and milk fat content (20,518, 10,074 and 11,518, respectively), while in the beef Charolaise breed, the largest number of candidate variants was associated with muscularity score measured at month 30 (17,044) (Supplementary Fig. 1).

The analysis revealed several genomic regions exhibiting highly significant effects on various traits and/or breeds. Notably, two regions stand out in particular. The first region, located on chromosome 14, approximately from positions 400,000 to 700,000, exhibits highly significant effects on milk composition and, to a lesser extent, on milk production across all three breeds in which these traits were measured, namely Holstein, Montbéliarde, and Normande. The second notable region, located at ~600,000 on chromosome 2, shows

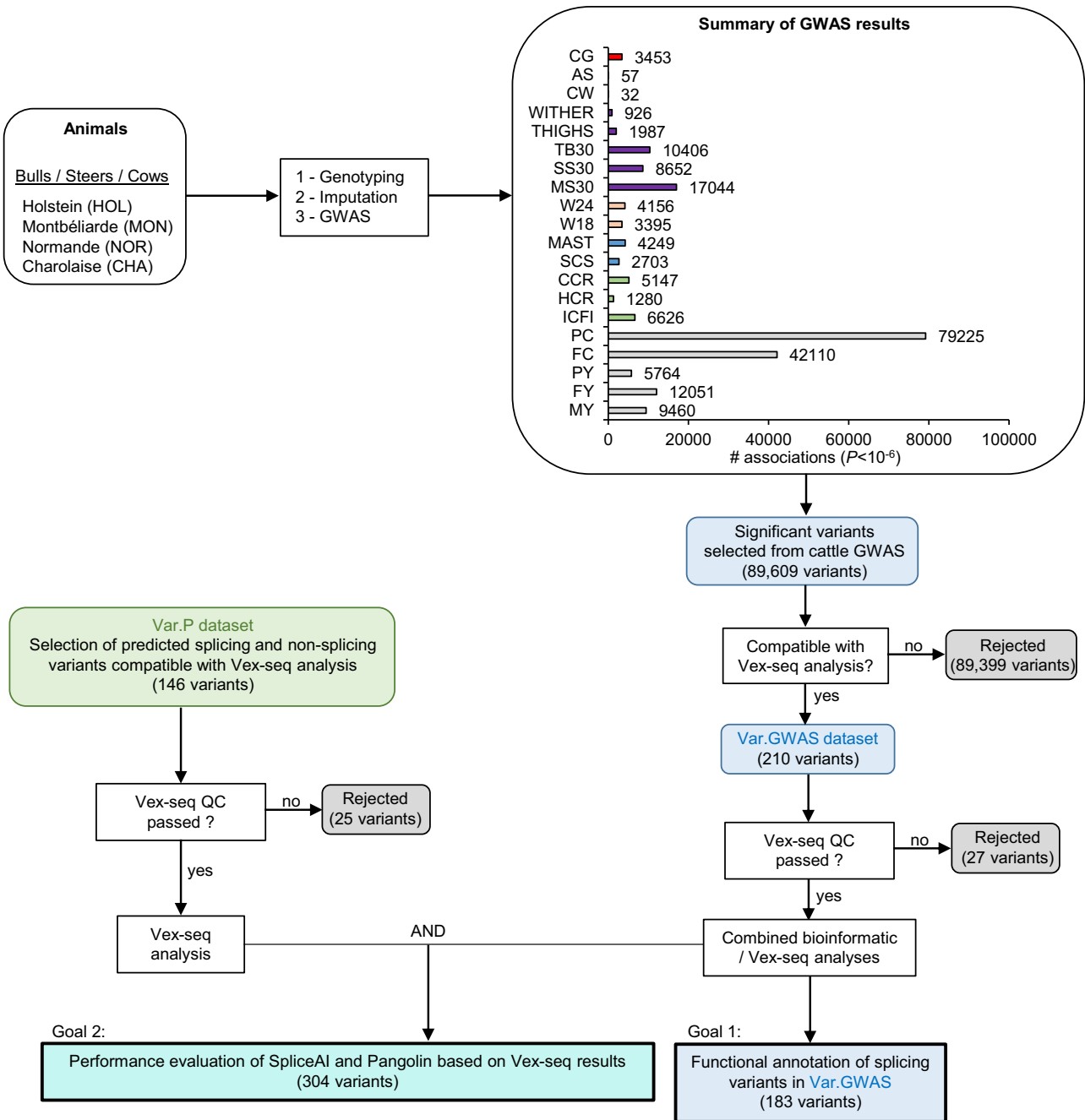

**Fig. 1 | Strategy used to identify SDV from bovine GWAS and to assess the performance of splicing prediction programs.** GWAS were performed on multiple agronomic traits in cattle, as described in the main text and Supplementary Data. A dataset (Var.GWAS) containing significant variants from GWAS was built up and functionally analysed to identify causal variants in the first instance. In parallel, an additional dataset (Var.P) containing arbitrarily selected variants with predictive splicing scores was used to increase the number of variants analysed by both in silico and Vex-seq methods in order to improve the performance evaluation of prediction programs.

significant associations with live or carcass morphology traits, specifically in the Charolaise and Normande breeds. It is worth noting that numerous other regions of interest were also identified.

## Use of SpliceAI and Pangolin in the context of the bovine genome

SpliceAI and Pangolin are deep-learning splicing prediction algorithms developed to analyse human genetic variants. SpliceAI has been constructed using as input only the genomic sequence of human pre-mRNA transcripts[60]. It provides a score reflecting the probability that a given variant increases or decreases the efficiency of splice sites located in its vicinity. Pangolin was constructed using sequence and RNA splicing measurements from four tissues (i.e. heart, liver, brain and testis) related to four species (i.e. human, rhesus, mouse and rat)[61]. Like SpliceAI, it provides a probability that a given variant impacts splicing. It has been established that the splicing code is highly conserved between mammals[62–64] and, thus, between humans and cows. This idea is supported by the similar observed frequency of nucleotides positioned close to the splice sites in both species, as illustrated by a sequence alignment of their splice regions (Supplementary Fig. 2). Thus, an algorithm designed to predict SDV in humans should also work efficiently in cattle.

Nonetheless, before making predictions on bovine candidate SDV using SpliceAI and Pangolin, we assessed the sensitivity of these

**Table 1 | Features of traits included in GWAS in the different breeds**

| Group of traits | Trait | Abbreviation | Holstein | Montbéliarde | Normande | Charolaise |
|---|---|---|---|---|---|---|
| Milk production | Milk yield | MY | 10,042 bulls | 3155 bulls | 2693 bulls | - |
| Milk production | Fat yield | FY | 10,042 bulls | 3155 bulls | 2693 bulls | - |
| Milk production | Protein yield | PY | 10,042 bulls | 3155 bulls | 2693 bulls | - |
| Milk production | Fat content | FC | 10,042 bulls | 3155 bulls | 2693 bulls | - |
| Milk production | Protein content | PC | 10,042 bulls | 3155 bulls | 2693 bulls | - |
| Fertility | Interval between calving and first artificial insemination | ICFI | 9718 bulls | 3012 bulls | 2603 bulls | - |
| Fertility | Heifers' conception rate | HCR | 9718 bulls | 3012 bulls | 2603 bulls | - |
| Fertility | Lactating cows' conception rate | CCR | 9718 bulls | 3012 bulls | 2603 bulls | - |
| Mastitis resistance | Somatic cell score | SCS | 10,066 bulls | 3115 bulls | 2697 bulls | - |
| Mastitis resistance | Clinical mastitis | MAST | 8792 bulls | 2724 bulls | 2255 bulls | - |
| Growth | Weight at 18 months | W18 | - | - | - | 7999 cows |
| Growth | Weight at 24 months | W24 | - | - | - | 7999 cows |
| Live morphology | Muscularity score at 30 months | MS30 | - | - | - | 8501 cows |
| Live morphology | Skeletal score at 30 months | SS30 | - | - | - | 8501 cows |
| Live morphology | Thickness of bones at 30 months | TB30 | | | | 8501 cows |
| Live morphology | Muscularity of the thighs | THIGHS | - | 3107 bulls | 2617 bulls | - |
| Live morphology | Muscularity of the withers | WITHER | - | 3107 bulls | 2617 bulls | - |
| Carcass | Carcass weight | CW | - | 4163 steers | 2730 steers | 4354 steers |
| Carcass | Age at slaughter | AS | - | 4163 steers | 2730 steers | 4354 steers |
| Carcass | Carcass grade | CG | - | 4163 steers | 2730 steers | 4354 steers |

programs, specifically on cattle, by scanning a positive set of SDV described in vivo and associated with phenotypes. A review of the literature and the OMIA database[65] allowed us to collect 24 bovine SDV supported by in vivo evidence, which we analysed with SpliceAI and Pangolin (Fig. 2a, b and Supplementary Data 1). To compare the performances of both programs, high recall (score ≥0.2), recommended (score ≥0.5), and high precision (score ≥0.8) thresholds were used as initially done to characterise SpliceAI[60]. As Pangolin and SpliceAI scores represent a probability for a variant to be an SDV, selecting a specific score threshold allows us to modulate the sensitivity and specificity of the prediction. SpliceAI scores range from 0 to 1 and are classified into four splicing effect categories depending on their predicted effect on the strength of the indicated splice site, which are (i) acceptor (AL) or (ii) donor loss (DL) and (iii) acceptor (AG) or (iv) donor (DG) gain. Pangolin scores range from −1 to 1 and can be interpreted as a probability to modify splicing where negative scores signify a decrease in the strength of the indicated splice site, and positive scores signify an increase in the strength of the indicated splice site. You will note that to allow comparison between SpliceAI and Pangolin results, Pangolin scores are sometimes noted in absolute values in this manuscript, and SpliceAI scores for the AL and DL classes are sometimes noted in opposite numbers (negative values). This is specified in the figures and their legends.

SpliceAI and Pangolin both returned a positive rate of predicted SDV of 70.8% (17/24) using the high recall threshold. With regard to the consequences of SDV on gene sequence, variants modifying the canonical splice site (CSSV) were predicted to be spliceogenic with 100% accuracy, whereas intronic variants were predicted incorrectly in more than half the cases (Fig. 2c). Taking into consideration the complexity of the affected phenotypes, 88.2% (15/17) of the variants responsible for monogenic diseases are predicted to be spliceogenic whereas only 28.6% (2/7) of variants involved in complex traits are predicted to be spliceogenic (Fig. 2d).

### Definition of Var.GWAS and Var.P datasets
Two variant datasets were generated to simultaneously identify causal SDV from bovine GWAS and to accurately assess the performance of the SpliceAI and Pangolin prediction tools for bovine variants (Fig. 1).

The Var.GWAS dataset contained all 210 significant GWAS variants fitting the constraints imposed by the design of our Vex-seq assay (Supplementary Fig. 3a). With our settings, technical limitations of the method only allowed the analysis of exons of 98 nt or less in length and their flanking introns, 50 nt upstream and 20 nt downstream. The first and last exons of genes and test sequences containing an *Mfe*I or *Spe*I restriction site were also excluded. Only significant GWAS variants (−log10(p value) >6) with a variant identifier (rsID) and highest imputation accuracy ($r^2 > 0.4$; mean($r^2$) = 0.827) were used to facilitate tracking variants through assembly, database browsing and ensure accuracy of the analysis. The Var.P dataset has been constructed from a pool of 1000 random bovine variants fitting Vex-seq constraints, subsequently analysed by SpliceAI, and filtered to keep only 146 variants with a balanced distribution of positive and negative splicing prediction scores. This second dataset, intentionally enriched with putative spliceogenic variants, has been created to increase the amount of putative SDV available to calculate SpliceAI and Pangolin performances. The identification of causal variants responsible for phenotype variation was performed solely using the Var.GWAS dataset, whereas both datasets were merged to calculate the performance of prediction tools.

### Vex-seq quality control (QC) and reliability
The Vex-seq analysis was performed using HEK293T and MAC-T cells according to previous studies and with minor modifications (Supplementary Fig. 3b, c)[57,66]. To ensure the production of high-quality data, several sequence quality filters were applied to remove reads that were not exploitable in downstream statistical analyses. The first step of the QC consisted of verification of the integrity of plasmid libraries (Supplementary Fig. 4a, b). Two criteria were selected to validate whether or not to keep a barcode (BC) for further analysis: no mutation in the BC sequence and at least 85% of correct reads from the MiSeq run for the associated test sequence. This resulted in the validation of 94.03% of BC. In a second step, variants associated with BC with too low expression (<10 reads) in the transcripts analysis were eliminated (Supplementary Fig. 4c). Finally, variants with less than 2 validated BCs for one of their alleles, or showing no splicing event neither for the reference (REF) nor for the alternative (ALT) allele were removed. Of

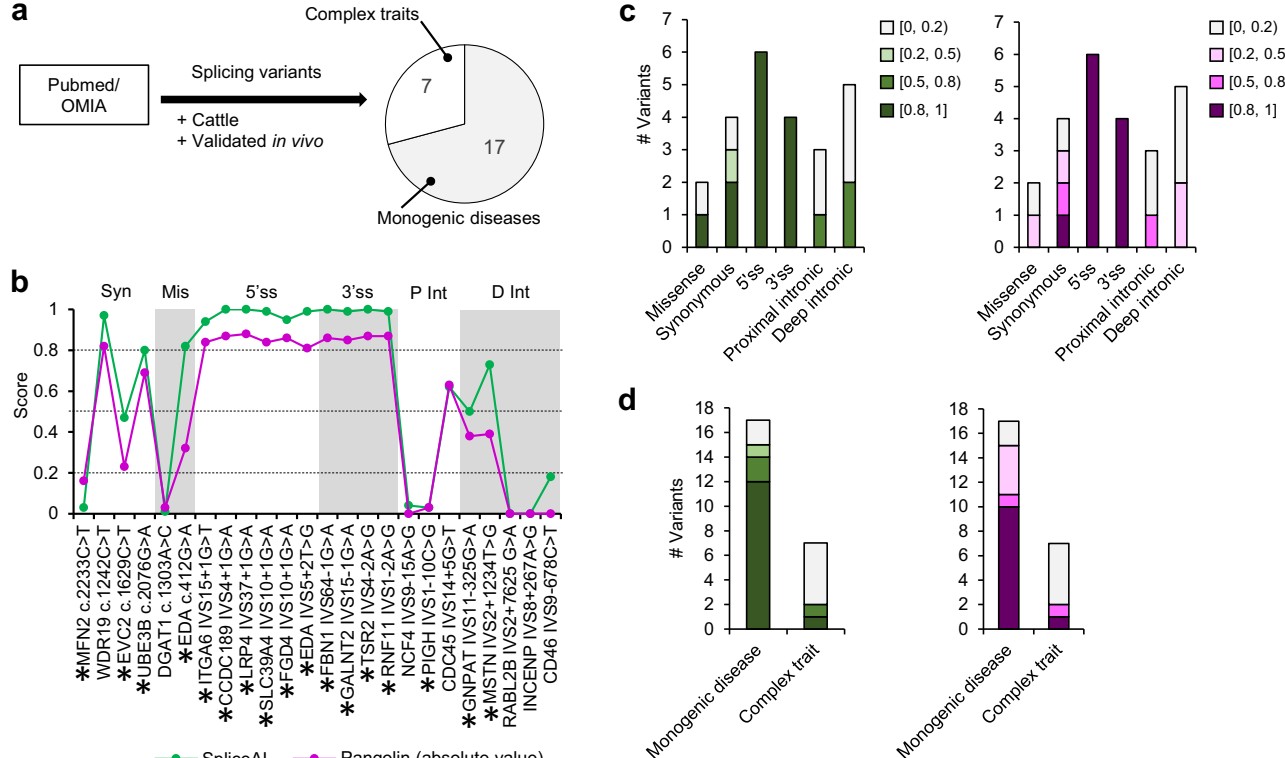

**Fig. 2 | SpliceAI and Pangolin programs predict bona fide SDV in cattle. a** A positive set of 24 SDV previously described in vivo and associated with phenotypes was compiled from PubMed and OMIA databases. **b** SpliceAI and Pangolin scores representing the probability to disrupt splicing were calculated for each variant classified by variant consequence. Variants with a score ≥0.2 (high recall threshold), ≥0.5 (recommended threshold), or ≥0.8 (high precision threshold) were predicted to be SDV as proposed by ref. [60]. Variants reported to be associated with a monogenic disease are marked by an asterisk; others are related to complex traits. Note that Pangolin scores are represented in absolute values to allow comparison with SpliceAI scores. Syn synonymous, Mis missense, 3'ss 3' splice site, 5'ss 5' splice site, P Int proximal intronic, D Int distal intronic. **c** Proportion of variants predicted to be spliceogenic or non-spliceogenic classified by variant consequence or **d** phenotype. Graphs displaying stacked green and purple bars indicate SpliceAI and Pangolin predictions, respectively. The number of variants for each score category is shown on the Y-axis, with the highest categories shown in the darkest colours. Variants with a score equal to or higher than 0.2 are predicted to be SDV. Pangolin scores are represented in absolute values. Source data are provided as a Source Data file.

the 919 variants analysed simultaneously using the Vex-seq method, approximately 75% of variants were validated in HEK293T or MAC-T cells, with a large overlap between cell lines (Supplementary Fig. 4d, e). However, it should be noted that the percentage of test exons that could be analysed was about 84% in both cell lines. This difference can be explained by the fact that, of the 919 variants initially included in the whole Vex-seq plasmid library, 99 are located in exon 8 of the *DGAT1* gene, and the majority of them did not pass the QC because exon 8 was not spliced in most cases. These 99 variants are not related to the Var.GWAS or Var.P datasets (see Materials and methods for details), but their failure to pass QC filters reduces the overall rate of variants validated for the whole Vex-seq analysis. If we focus specifically on datasets presented in this study, 87.1% (183/210) and 82.9% (121/146) of variants were validated for Var.GWAS and Var.P datasets, respectively.

Read counts associated with each BC were used to calculate the percent spliced-in (PSI) index. A high reproducibility between BC replicates was observed within a given cell type ($r > 0.96$ for both HEK293T and MAC-T cells) which indicates that the BC sequence had little influence on the splicing of the test exon (Supplementary Fig. 5a). A comparison of PSI between biological replicates showed a good correlation considering triplicate within each cell line ($r > 0.94$ and $>0.92$ for HEK293T and MAC-T cells, respectively) and also, albeit to a lesser extent, between cell lines ($r > 0.86$) (Supplementary Fig. 5b).

Finally, we verified that Vex-seq results reflect biological reality using a set of bona fide bovine and human SDV, which were also included in the Vex-seq analysis. SDV reported to be highly

spliceogenic in cattle ($n = 2$) and in humans ($n = 5$) were tested and yielded similar splicing behaviour in vivo and in the Vex-seq analysis (Supplementary Fig. 6 and Supplementary Data 2). Moreover, 13 SDV exclusively localised in human *ABCA4* gene previously analysed by means of midigene assays were also included in the analysis[67]. These midigenes were splice vectors of varying lengths (up to 11.7 kb) covering almost the entire *ABCA4* gene and transfected in HEK293T cells, which allowed investigation of the effect of SDV in a relatively large sequence context. This additional group of variants with various functional impacts enabled the performance of Vex-seq to be calculated (Supplementary Fig. 7 and Supplementary Data 2). The dataset from which these 13 variants were extracted has previously been used as a benchmark to evaluate the performance of splicing prediction programs, which is why we chose to use it[68]. Vex-seq ΔPSI values were ranging from −9 to −88 and were highly correlated between cell lines (Supplementary Fig. 7a–c). In order to allow comparison between Vex-seq analysis outputs (PSI and ΔPSI) and midigene analysis outputs from the study by Sangermano et al. (% of abnormal *ABCA4* transcripts associated with the ALT variant allele)[67], we converted PSI values of REF and ALT alleles into a % of abnormal *ABCA4* transcripts associated with ALT allele according to the following formula: 100 − ([PSI(ALT)/PSI(REF)])*100. The comparison of this percentage for each variant obtained with Vex-seq against midigene showed both approaches yielded similar outcomes with a Pearson correlation coefficient of 0.7891 and 0.8968 in HEK293T and MAC-T cells, respectively (Supplementary Fig. 7d). Vex-seq sensitivity and specificity were calculated using midigene data as benchmark. Sensitivity was between 95.45 and

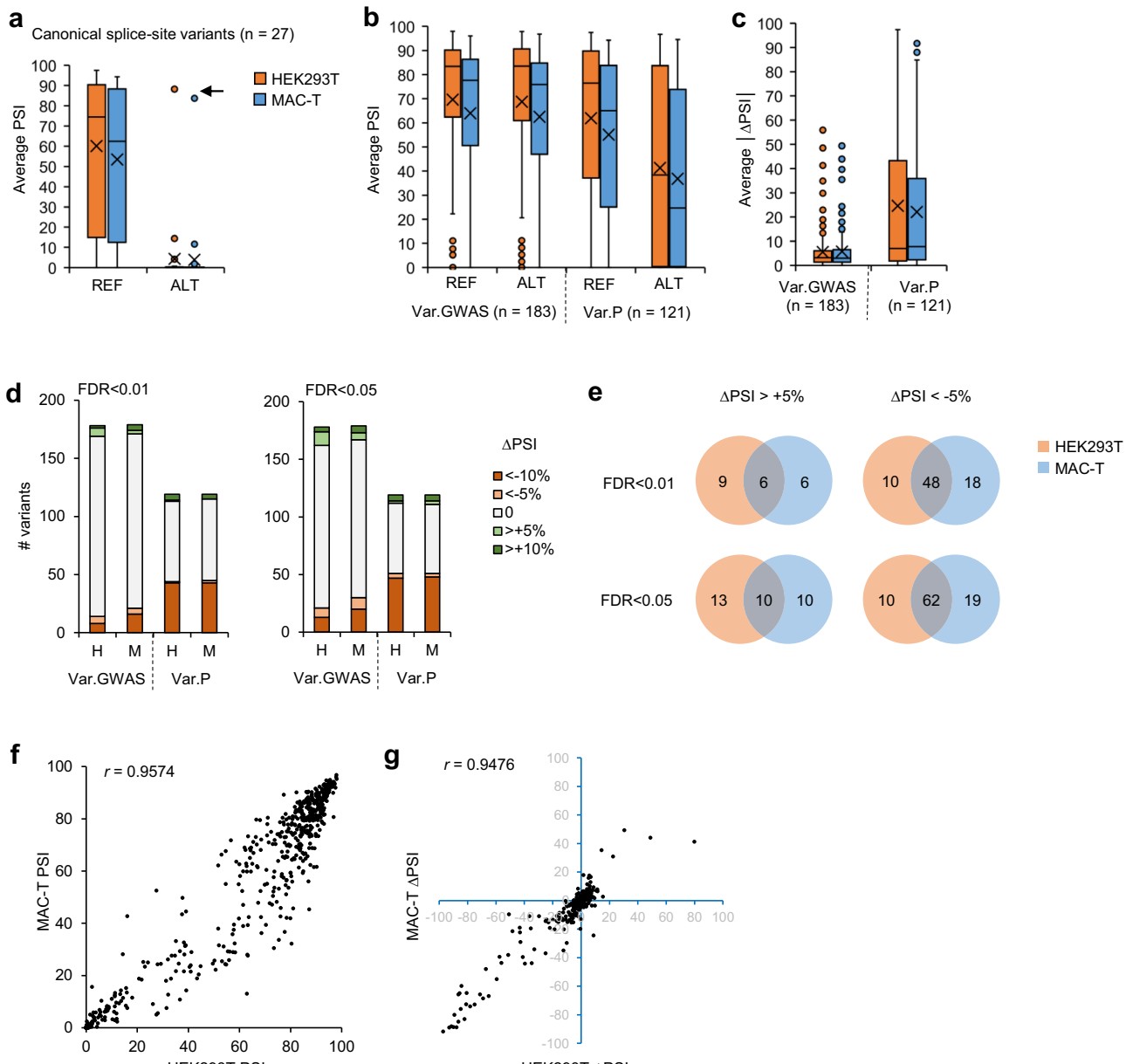

**Fig. 3 | Summary of Vex-seq analysis. a** Average PSI for reference and alternative alleles of the 27 CSSV that are part of the Var.GWAS and Var.P datasets. Of note, one CSSV (rs448758869) resulting in a GT > GC transition still allowed the test exon to be included efficiently (arrow). Box plots are defined as follow; cross, mean; centre line, median; box limits, upper and lower quartiles; top and bottom whisker lines, maximum and minimum values; points, outliers. **b** Average PSI for reference and alternative alleles of variants from whole Var.GWAS and Var.P datasets. Box plots are defined as in (**a**). **c** Average │ΔPSI│ variant by dataset. Box plots are defined as in (**a**). **d** Proportion of tested variants confirmed to be SDV determined for each dataset. Variants with ΔPSI above 5% or below 5% were considered to be SDV. FDR false discovery rate used to identify SDV. H HEK293T, M MAC-T. **e** Overlap of SDV from merged Var.GWAS and Var.P datasets between HEK293T and MAC-T cell lines. Scatter plot of **f** PSI and **g** ΔPSI values from merged Var.GWAS and Var.P datasets in HEK293T versus MAC-T cells. *r* Pearson correlation coefficient. Source data are provided as a Source Data file.

100%, and specificity was between 50 and 75% (Supplementary Fig. 7e, f). However, this result should be treated with caution as 13 samples is low to estimate these parameters.

**Vex-seq results summary**
In eukaryotes, the composition of canonical 3′ splice sites (3′ss) and 5′ splice sites (5′ss) is almost exclusively AG and GT, respectively[69,70]. Rarely, a GC dinucleotide is observed for the 5′ss[69]. The consequence is that a nucleotide change in a 3′ss or 5′ss inevitably leads to a loss of function of these splice sites, except in the case of a + 2 T > C transition where the function of the 5′ss may be preserved[71]. Considering the Var.GWAS and Var.P datasets together, there were 27 CSSV presenting

a REF allele average PSI of around 55% in both cell lines. Introducing the ALT allele within the sequence led to a dramatic decrease of average PSI with the exception of rs448758869, which was a + 2 T > C transition (Fig. 3a). PSI and ΔPSI were then calculated for the entire Var.GWAS and Var.P datasets which revealed that average PSI values were globally slightly lower in MAC-T cells compared to HEK293T cells (Fig. 3b). Also, average PSI values for the ALT allele were approximately two-thirds of those for the REF allele in the Var.P dataset, whereas no obvious difference was observed on this point in the Var.GWAS dataset. This resulted in an average |ΔPSI| of around 25% for Var.P, whereas only several extreme values with a |ΔPSI| above 10% were observed in the Var.GWAS dataset (Fig. 3c).

To statistically sort spliceogenic variants from non-spliceogenic ones in each dataset, we defined a threshold of ±5% for the ΔPSI, associated with a false discovery rate (FDR) of less than 0.05 or 0.01, to test different stringencies (Fig. 3d). Using an FDR <0.01, 12.9% (23/178) and 16.2% (29/179) of variants from Var.GWAS were classified as SDV in HEK293T and MAC-T cells, respectively, and 42% (50/119) and 41.2% (49/119) of variants from Var.P were classified as SDV in HEK293T and MAC-T cells, respectively. Increasing the FDR threshold to 0.05 slightly increased these proportions (Fig. 3d, right). A predominance of SDV decreasing rather than increasing test exon inclusion was observed, especially in the Var.P dataset. The majority of SDV was identified in both cell lines (Fig. 3e). Finally, a strong correlation was observed between the two cell lines for PSI and ΔPSI, with Pearson correlation coefficients of 0.9574 and 0.9476, respectively (Fig. 3f, g). All the information about variants of the Var.GWAS and Var.P datasets are available in Supplementary Data 3, 4.

## Performance evaluation of bioinformatic programs

Prediction scores related to variants from Var.GWAS were generated using SpliceAI and Pangolin, as previously done for variants constituting Var.P (Fig. 4a). Only 2.2% (4/183) and 1.64% (3/183) of variants from Var.GWAS were predicted to be spliceogenic by SpliceAI and Pangolin using the high recall threshold of 0.2, respectively. The results obtained on the two datasets showed a high correlation between the predictions of the two programs ($r = 0.8982$) (Fig. 4b). On the other hand, ΔPSI values and prediction scores were weakly correlated with a little higher $r$ for Pangolin (Fig. 4c). Also, several positive prediction scores were found for variants which yielded negative ΔPSI values of variants. In fact, the creation of a cryptic splicing site is predicted as a splicing gain by SpliceAI and Pangolin, which is logical, but the consequence from a functional point of view is the non-use of the canonical splicing site, which results in the exclusion of the test exon and therefore a reduction of the ΔPSI. SDV identified in Var.GWAS and Var.P datasets using Vex-seq analysis were used to calculate the true positive (TP) rate, and non-SDV to calculate the true negative (TN) rate of SpliceAI and Pangolin by score range using the threshold 0.2. Pangolin showed a better TP rate than SpliceAI in each score range. Also, the FDR <0.01 condition yielded more reproducible results between cell lines (Fig. 4d). Then, ROC curves were generated for each programme, and AUC was calculated. This revealed that Pangolin (average AUC = 0.831) slightly outperformed SpliceAI (average AUC = 0.775) (Fig. 4e).

## Using SpliceAI and Pangolin low thresholds to filter SDV involved in complex traits

Interpretation of SDV effects using splice predictors requires the choice of a pre-determined score threshold beyond which variants are considered spliceogenic. A score threshold of 0.2 has been proposed by the authors of SpliceAI and Pangolin to make permissive predictions[60,61]. In the context of searching for causal variants identified in bovine GWAS, and depending on the strategy chosen, it may be useful to lower the threshold of SpliceAI and Pangolin to increase the number of positive hits and, thus, the number of candidate variants to be investigated in downstream validation experiments. In this case, the optimal threshold is usually lower than those recommended by authors and may depend on the composition of the variant dataset to be analysed[54]. We calculated TP and FP rates in the Var.GWAS dataset for thresholds below 0.2 (Fig. 5a). For example, lowering the threshold to 0.1 retrieves 11 variants predicted to be spliceogenic, including 6 false positives according to Vex-seq data performed on HEK293T cells. Although this increases the total number of variants to be functionally assessed (11 instead of 3 when the threshold used is 0.2), it will lead to the identification of 2 additional SDV. Here too, Pangolin performed better than SpliceAI by using thresholds below 0.2 (Fig. 5b).

## Phylogenetic conservation, sequence features and genetic data associated with spliceogenicity

Phylogenetic conservation of nucleotides affected by variants from Var.GWAS and Var.P datasets were assessed using the GERP score[72]. Scatter plots between ΔPSI values and GERP scores showed significant enrichment of variants affecting phylogenetically conserved nucleotides within the group of variants decreasing splicing rate (Fig. 6a, c). This was statistically confirmed by an enrichment test which exhibited a -1.8-fold increase (two-tailed Fisher's exact test; $p$ value <0.0001) of conserved variants in the "Loss" category compared to 'Neutral', observed in both cell lines (Fig. 6b, d). This trend was also observed between splicing prediction scores and GERP scores (Fig. 6e, g) with an enrichment of 2.49- ($p$ value <0.0001) and 2.26-fold ($p$ value <0.0001) for SpliceAI and Pangolin, respectively (Fig. 6f, h). The variant consequences were characterised using Variant Effect Predictor (VEP)[73] on canonical and alternative transcripts[73]. A small fraction of variants impacting canonical splice donor sites were not predicted as SDV by Pangolin because it ran only on canonical transcripts (Fig. 6i–l). Also, some SDV that impact splice donor sites were not validated as spliceogenic by Vex-seq. This is because their PSI is less than 5% for the REF allele, and therefore the ΔPSI cannot reach the 5% threshold needed to classify the variant as an SDV, even in the case where the ALT allele completely disrupts splicing. Next, we found that GWAS variants associated with the most severe splicing alterations also seemed to be those with the lowest frequency for the ALT allele, both in terms of ΔPSI or predictive scores (Fig. 7a, b). In addition, among the 182 QTL related to the phenotypes studied here, 22 (11.2%) contained at least one SDV (noted QTL+) (Fig. 7c). This is an underestimation of the total number of SDV lying in QTL mainly because only a relatively small fraction of the candidate SDV could be analysed using Vex-seq due to methodological constraints. The Vex-seq regions ($n = 54,149$) represent 22.5% of all exonic regions in the bovine genome, and only 183/210 GWAS variants detected in these regions have been analysed after applying the Vex-seq QC filters. This indicates that the actual proportion of QTL+ among QTL identified in this study can be estimated to be 5.04 times greater. This means that QTL+ corresponds to 57% of all QTL ($11.2 \times (100/22.5) \times (210/183)$) (Fig. 7d). It should be noted that this number is purely indicative and that the actual proportion of QTL+ may be slightly different. In fact, the spliceogenic properties of exons may vary according to their length and position, and the causal variants are not distributed in a strictly equal manner in the QTL.

## Identification of causal SDV

The 38 SDV identified in the Var.GWAS dataset (Table 2 and associated traits in Supplementary Data 5) were assessed for their consequence on the transcript primary sequence (Fig. 8a). This information was used to predict the deleterious effect expected at the protein level, allowing in fine to functionally classify each of the two alleles for each variant. Three lines of evidence were thus used to filter and validate causal variants (summarised in Fig. 8b): (i) a statistical association between the variant and the phenotype; (ii) a predicted functional impact of the variant on the gene function (i.e. a gain or a loss of protein expression/function resulting from the splicing disruption); and (iii) a functional link between the gene and the considered phenotype. The last point can be documented by direct experimental demonstration using rare natural variants with large effect sizes identified in cattle[45], or indirectly inferred by connecting multiple scientific evidences together to reconstitute the biological process involved in the gene-phenotype relationship. Based on these criteria, three putative causal variants were identified, namely rs134725785, rs135835897, and rs133242826, located in *DGAT1*, *PIK3C2G* and *PIAS4*, respectively (Fig. 8b, c). The effects of these 3 variants on mRNA transcripts and protein sequences with predicted functional impact are detailed in Supplementary Fig. 8.

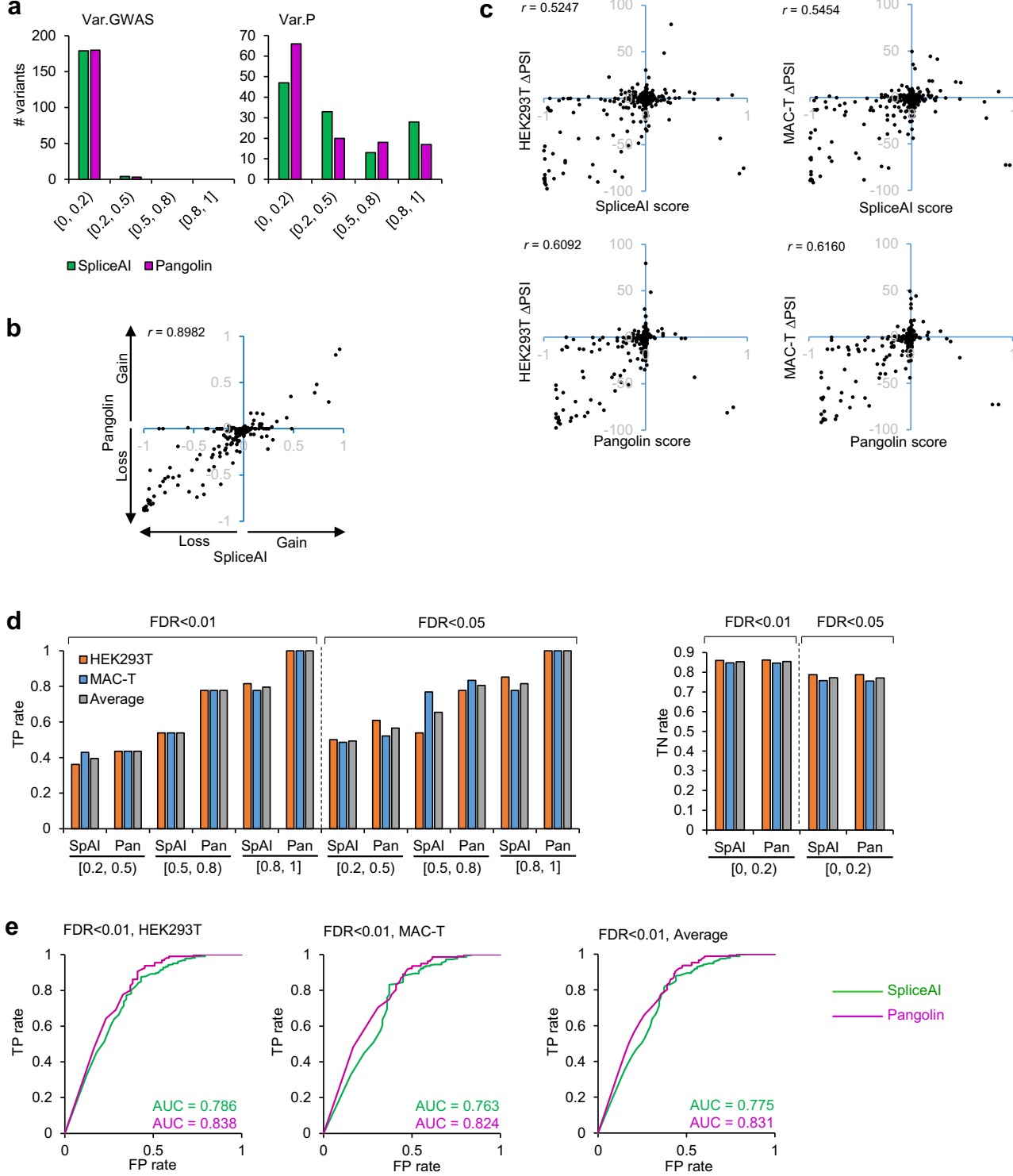

**Fig. 4 | Performance evaluation of SpliceAI and Pangolin using merged Var.G-WAS and Var.P datasets. a** Distribution of spliceAI and Pangolin absolute scores in Var.GWAS and Var.P datasets. **b** Scatter plot of SpliceAI versus Pangolin scores. SpliceAI scores related to donor or acceptor loss have been turned into opposite numbers to allow comparison with Pangolin scores. *r* Pearson correlation coefficient. **c** Scatter plot of SpliceAI and Pangolin scores each versus ΔPSI values in both HEK293T or MAC-T cell lines. **d** True positive and true negative rates of SpliceAI and Pangolin obtained and classified by absolute score category. SDV identified using Vex-seq were used to calculate the true positive rate, and non-SDV to calculate the true negative rate. TP true positive, TN true negative, SpAI SpliceAI, Pan Pangolin. **e** Receiving operating curve (ROC) and area under the curve (AUC) were calculated for SpliceAI and Pangolin in each cell line and for the mean values between the two cell lines. Source data are provided as a Source Data file.

## Colocalization between SDV from Var.GWAS and eQTL/sQTL SNP

We searched for shared SNP between the 38 SDV from Var.GWAS and eQTL/sQTL data, through a review of published studies reporting both eQTL and sQTL SNP. Four studies did not allow the identification of shared SNP[74–77]. Two studies conducted on imputed WGS and involving large numbers of animals and samples showed positive results with respect to single-tissue eQTL/sQTL SNP[7,25] and multi-tissue eQTL/

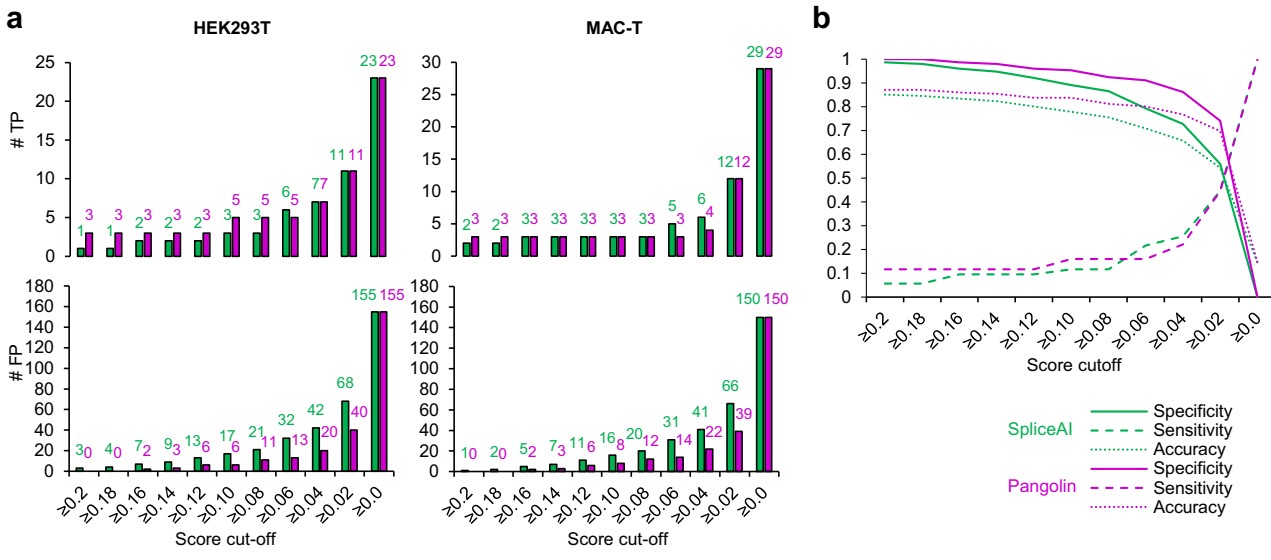

**Fig. 5 | Determining optimal score threshold in the Var.GWAS dataset. a** True positive and true negative rates, in addition to **b** sensitivity, specificity and accuracy of SpliceAI and Pangolin, are shown for thresholds below 0.2. Note that Pangolin scores are represented in absolute values to allow comparison with SpliceAI scores. Source data are provided as a Source Data file.

sQTL SNP[25] (Supplementary Data 6). SDV colocalize with six single-tissue eQTL SNP and eight single-tissue sQTL SNP. Multi-tissue analysis yielded better results, indeed, SDV also colocalized with 23 multi-tissue eQTL SNP and 23 multi-tissue sQTL SNP. Globally, 71.1% of SDV (27/38) colocalize with at least one single- or multi- e/sQTL SNP.

## Discussion

We set out to explore the impact of SDV in complex traits in cattle using the latest generation of computational and experimental tools. To do this, we integrated GWAS, MPRA and in silico splicing prediction data in an original approach aimed at drawing the most biologically relevant conclusions as possible. Thanks to this, we identified three putative causal SDV involved in production traits in cattle.

GWAS were conducted on large populations of bovine animals with accurate phenotypic records and at the sequence level, which corresponds to the best possible resolution for GWAS. These analyses resulted in the detection of 573 QTL associated with 20 various traits of interest. Many of these QTL overlap with regions already reported in CattleQTLdb (https://www.animalgenome.org/cattle)[78]. However, only a limited number of causal variants within these regions have been previously identified. Notable examples include the missense causal variant within the *DGAT1* gene on chromosome 14, which exerts a strong influence on milk composition[79], and the various missense variants within the *MSTN* gene, known for their major impact on animal muscularity[80]. The limited number of identified causal mutations highlights the considerable challenge associated with their discovery, emphasising the importance of having efficient methods for their detection.

The Vex-seq method had previously been used to analyse human SDV[57], and we have carried it out on livestock species in the present study. The results we obtained are overall comparable to those of ref. 57 although we made some minor modifications to the protocol, such as adding an additional BC per construct or using different mammalian cell lines. Consistent with their observations in humans, we found that the PSI varied greatly based on the sequence of the test exons, ranging from 0 to 98%. Thus, it should be noted that some variants could not be analysed due to their location in regions in which test exons were not included. This phenomenon shows that not all exons can be analysed in Vex-seq, and that in some cases a wider sequence context than those of a minigene is probably required to analyse the impact of SDV[81,82]. Despite this, our Vex-seq analysis gave biologically relevant results, validated by a multitude of positive controls and QC steps. MPRA for splicing is time-consuming, costly, and technically complex, and for this reason, only one or two cell lines are usually employed at the same time[55,57,58]. We decided to use one bovine cell line and one human cell line already used as models to study splicing in these two species. HEK293T cells have been very frequently used to perform low-throughput splicing assays (see examples in human[83,84] and cattle[46,85]) but also in the context of the two other MPRA methods developed to analyse SDV[55,58]. In some cases, comparisons with in vivo data from patients have been done and yielded consistent results[55]. HEK293T cells have also been used to generate the *ABCA4* benchmark dataset we used to validate the Vex-seq[67]. MAC-T cells have been widely used to model epithelial cells of the mammary gland (for a recent example, see ref. 86). Since the majority of SNP identified in our GWAS concern milk traits, the choice of this cell line was considered to be the most appropriate. In addition, MAC-T cells have already been successfully used to develop splicing assays for bovine genes[85,87,88]. PSI and ΔPSI values both showed a strong correlation between the HEK293T and MAC-T cell lines, which suggests good phylogenetic conservation of splicing mechanisms between humans and cattle. It also highlights the predominantly ubiquitous nature of SDV as previously observed[57]. Only one variant showed an inverse effect in the two cell lines (*SULT1B1* rs208961079). Further functional characterisation of this SDV could lead to a better understanding of the impact of genetic alterations on the tissue-specific regulation of splicing, which remains a burning question[89,90]. According to cGTEx data, *SULT1B1* is mainly expressed in the digestive tract, the bone marrow, adipose tissues, sperm, and kidneys, but also in the adrenal gland and, to a lesser extent, in mammary tissues[7]. HEK293T cells transcriptome more closely resembled that of adrenal cells than kidney cells[91], and MAC-T is a model for epithelial cells of the mammary gland. Thus, the cell lines used to perform the Vex-seq analysis fit with the expression pattern of *SULT1B1* in two tissues, and further experiments to characterise its transcript isoforms in adrenal and mammary glands in animals carrying different genotypes for rs208961079 could validate the tissue-specific nature of this SDV. Finally, we confirmed that in cattle, as in

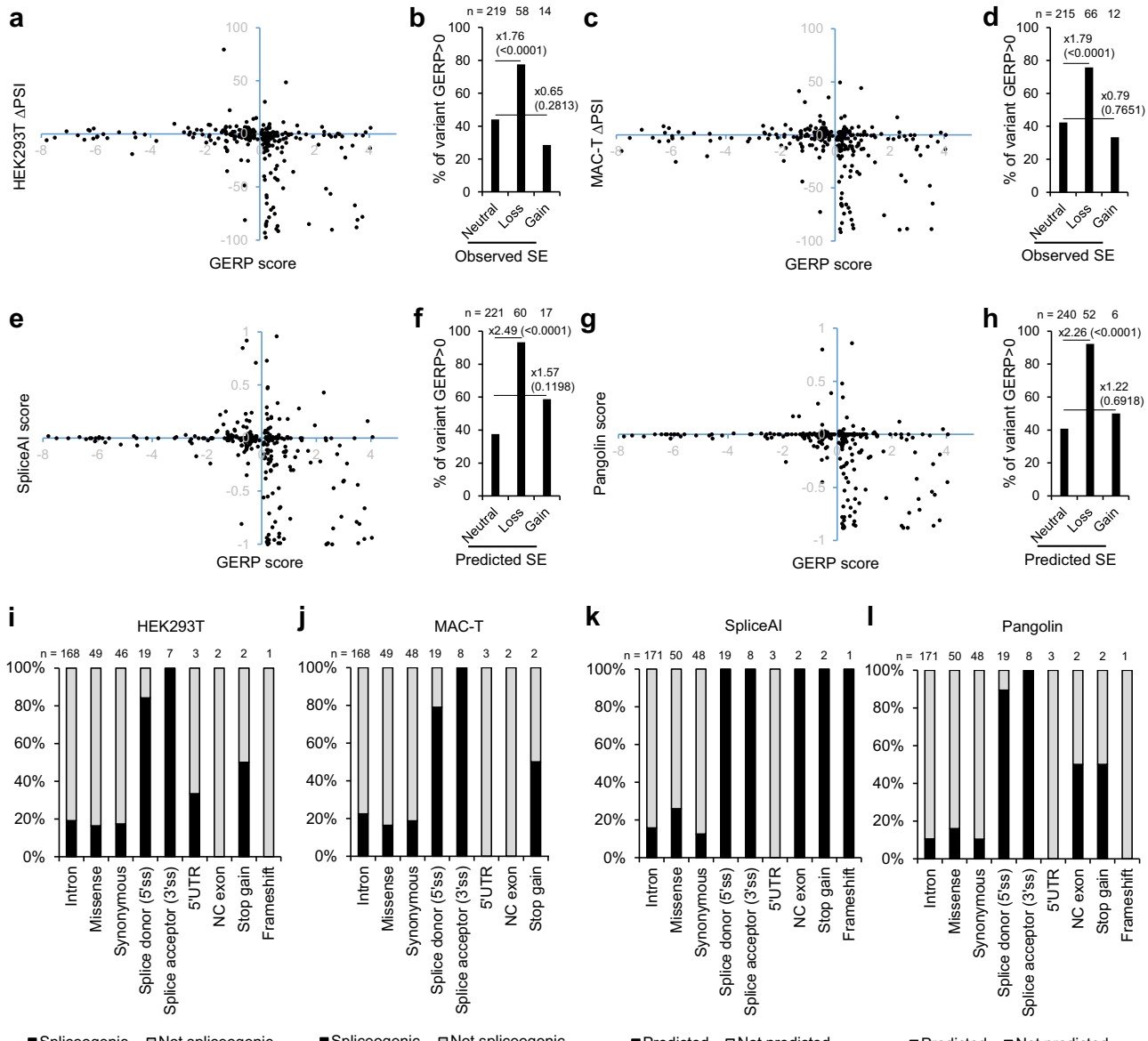

**Fig. 6 | Correlation between spliceogenicity and either phylogenetic conservation or variant consequence observed in the merged Var.GWAS and Var.P datasets.** Scatter plots of GERP conservation scores versus ΔPSI values in **a** HEK293T and **c** MAC-T cells or versus prediction scores of **e** SpliceAI and **g** Pangolin. The highest GERP scores represent the highest phylogenetic conservation[72,125]. Enrichment in GERP-positive variants within spliceogenic or non-spliceogenic variants as determined by Vex-seq in **b** HEK293T and **d** MAC-T cells and as predicted by **f** SpliceAI and **h** Pangolin. Statistical enrichment within the

'Loss' and 'Gain' categories compared to 'Neutral' was assessed using a two-tailed Fisher's exact test. The sample size in each category (*n*), the fold change and the *p* value (in brackets) are indicated. Bars indicate the percentage of GERP-positive variants for the three different categories of splicing effect (SE). Thresholds used to define categories are ±5% for ΔPSI values (FDR <0.01) and ±0.2 for prediction scores. Proportion of SDV in each variant consequence category as determined by Vex-seq in **i** HEK293T and **j** MAC-T cells or as predicted by **k** SpliceAI and **l** Pangolin. Source data are provided as a Source Data file.

humans, variants that alter conserved nucleotides are more likely to affect splicing[57].

Another substantial methodological contribution brought by our study relates to the use of SpliceAI and Pangolin splicing prediction programs in the context of the bovine genome. The composition of exon-intron boundaries, as well as the sequence of splice sites and splicing regulatory elements, is well conserved between mammals[62–64]. As SpliceAI and Pangolin have been trained on human data and predict SDV directly from the primary sequence of genes, we expected they would perform well in cattle. It was, however, necessary to validate their usefulness in this context using experimental data, which, at the same time, made it possible to accurately assess their performance. With respect to data obtained from in vivo studies, both programs were able to detect bovine SDV responsible for different monogenic

diseases with an overall sensitivity rate of 88.2% when applying a threshold of 0.2. This result was very close to the sensitivity of SpliceAI (89.9% using the same threshold) calculated from predictions made on human SDV identified in the context of the molecular diagnosis of genetic disorders[92]. To our knowledge, there is currently no similar data available for Pangolin. The analysis of the SDV associated with complex traits presented a low true positive rate (28.6%), probably because they are often hypomorphic and lead to less radical changes in splicing regulatory sequence features. A more in-depth performance evaluation of SpliceAI and Pangolin was then carried out using Vex-seq data, and this revealed several important points. First, Pangolin slightly outperformed SpliceAI for splicing prediction of bovine SDV, evidenced by the measurement of the AUC of ROC curves. This is in line with the observations made in the previous studies, which compared

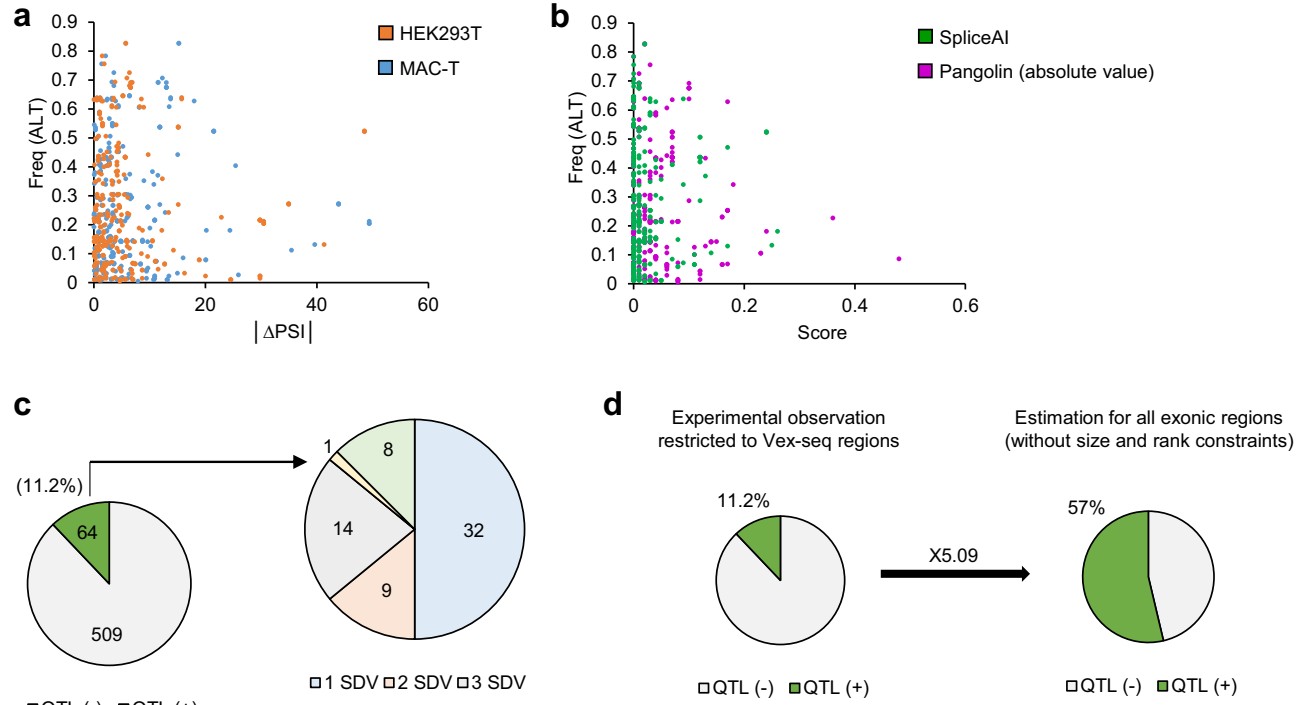

**Fig. 7 | Genetic features of SDV from the Var.GWAS dataset.** Scatter plots of **a** |ΔPSI| values or **b** prediction scores versus ALT allele frequency. Note that Pangolin scores are represented in absolute values to allow comparison with SpliceAI scores. **c** Proportion of QTL containing at least one SDV identified in Vex-seq

(QTL + ). The number of QTL+ containing more than 1 SDV is detailed on the right. **d** Based on experimental observations restricted to Vex-seq regions, an estimate of the proportion of QTL+ regions was made for the whole genome (see main text for details). Source data are provided as a Source Data file.

the performance of these two programs using human MPRA data from the literature[61,54]. The Pangolin algorithm was trained on transcriptomic data from four different mammals, which may also explain why it gave more accurate results on a non-human mammal, by contrast with SpliceAI, which was trained exclusively on human data. Furthermore, we addressed the issue of selecting the optimal threshold for these programs to effectively predict SDV in a GWAS dataset. This choice depends on both the algorithm used and the dataset analysed[54]. Based on human MPRA data and depending on the selected criteria, Smith and Kitzman determined a median optimal cutoff comprised between 0.8 and 0.14 for SpliceAI and equal to 0.12 for Pangolin[54]. In the context of bovine GWAS performed on complex traits, we observed that lowering the minimal recommended threshold of 0.2 and tolerating a higher number of false positives allow for an increase in the number of true positives SDV, which may hold potential value for further functional validation.

This study represents a large-scale analysis of SDV in farm species. Our results showed that SDV are widely associated with bovine phenotypes and are expected to be found in the majority of QTL (more than 50%), which suggests a pervasive role for this class of variants in the plasticity of complex traits in cattle. It should be noted that deep intronic variants have not been integrated into our analysis, which means that the role of SDV may be even more extensive than we estimated on the basis of our data. Our findings are also consistent with previous observations, which have found that multi-tissue cis-sQTL accounts for about 60% of the heritability of complex traits in cattle[25].

A total of 38 SDV identified from GWAS results were experimentally validated using the Vex-seq method. Despite they all alter splicing, it has not always been possible to clearly infer their effects on protein function. We considered that variant alleles that caused skipping of the test exon can be interpreted as 'loss-of-function' (LoF) when they led to the creation of a premature termination codon (PTC) located

upstream of the last exon. In fact, newly synthesised mRNAs that contain PTC are degraded through the NMD mechanism in order to prevent the production of truncated proteins[93]. We were able to classify 19 variants from GWAS as LoF or 'gain-of-function' (GoF), including one which suppressed the ATG translation start site. Molecular QTL data from two studies showed that 27 out of the 38 SDV colocalize with cis-sQTL and cis-eQTL SNP in cattle[7,25]. This finding supports that these variants are true SDV. It should be noted that these two studies led to the identification of colocalized SDV as they were carried out on a large number of animals and samples, allowing the identification of many molecular QTL. Also, the multi-tissue analyses are more powerful and identify more QTL than single-tissue ones, which also explains why they colocalized with a greater number of SDV. Ten SDV did not colocalize with any molecular QTL, but this can be explained in various ways. The consequences of an SDV may be undetectable by sQTL or eQTL analyses if it is not expressed in the tissues tested or at the right timing. The power and completeness of these QTL analyses is also limited by the size and genetic background of the samples available.

Finally, the functional validation of GWAS variants has enabled us to identify three putative causal variants that impact *DGAT1*, *PIK3C2G* and *PIAS4* gene function and are involved in complex phenotypes in cattle. First, one of these three putative causal SDV was rs134725785. It increased *DGAT1* expression as determined by Vex-seq, and was associated with variation in milk phenotypes in our GWAS. *DGAT1* encodes diacylglycerol O-Acyltransferase 1, an enzyme catalysing the final step of the biosynthesis process of triacylglycerol. Two *DGAT1* loss-of-function variants altering milk phenotypes in cattle have been reported to date, namely p.M435L[45] and p.K232A[79,94]. The p.M435L variant is a rare SDV that results in a non-functional truncated protein[45]. The well-known p.K232A is responsible for a significant decrease in cow's milk fat yield via an alteration in the enzymatic activity of DGAT1[79,94]. Furthermore, this variant and those in LD with it are associated with splicing changes in some introns of this gene[87,88].

**Table 2 | The 38 SDV from the Var.GWAS dataset validated using Vex-seq**

| Chr | Position | rsID | REF | ALT | Gene symbol | Breed[a] | F(ALT)[b] | ΔPSI(H) | ΔPSI(M) |
|---|---|---|---|---|---|---|---|---|---|
| 7 | 19984038 | rs133242826 | G | A | PIAS4 | CHA | 0.205 | 30.4 | 49.4 |
| 4 | 76710798 | rs385063917 | T | A | PPIA | NOR | 0.113 | N.S. | 35.4 |
| 3 | 15041705 | rs109482780 | C | A | MSTO1 | HOL | 0.629 | N.S. | 18.0 |
| 5 | 91787772 | rs135835897 | C | A | PIK3C2G | MON | 0.086 | N.S. | 16.0 |
| 18 | 44900183 | rs136188475 | A | G | WTIP | HOL | 0.146 | 10.5 | N.S. |
| 3 | 15541059 | rs135598974 | T | C | ADAM15 | HOL / MON | 0.044 / 0.015 | 8.6 | 9.4 |
| 7 | 17999133 | rs477089248 | G | A | DENND1C | CHA | 0.143 | N.S. | 8.7 |
| 3 | 16482608 | rs135482846 | T | A | CRTC2 | HOL | 0.126 | 6.0 | 10.2 |
| 14 | 687854 | rs135457327 | G | A | MROH1 | NOR | 0.066 | 7.4 | N.S. |
| 14 | 610004 | rs134725785 | G | A | DGAT1 | NOR | 0.068 | 7.0 | N.S. |
| 6 | 85424552 | rs137349211 | C | T | CSN1S1 | NOR | 0.290 | N.S. | 6.5 |
| 6 | 85537562 | rs108967182 | C | T | CSN1S2 | MON / NOR | 0.682 / 0.726 | 6.4 | N.S. |
| 15 | 77122984 | rs109702869 | T | C | ACP2 | CHA | 0.647 | 5.2 | N.S. |
| 20 | 33456383 | rs133343716 | C | T | MROH2B | HOL | 0.305 | N.S. | −6.1 |
| 14 | 446321 | rs450686754 | C | T | PPP1R16A | NOR | 0.095 | N.S. | −6.4 |
| 6 | 85428068 | rs133474041 | G | A | CSN1S1 | NOR | 0.295 | N.S. | -6.8 |
| 6 | 85199301 | rs208961079 | C | T | SULT1B1 | NOR | 0.181 | 8.7 | −24.4 |
| 6 | 83318905 | rs209655753 | G | A | STAP1 | NOR | 0.189 | −8.1 | N.S. |
| 5 | 119386807 | rs379016630 | G | T | IL17REL | MON | 0.143 | −7.3 | −9.6 |
| 2 | 1268402 | rs382712574 | G | A | TUBGCP5 | CHA | 0.101 | −8.5 | N.S. |
| 6 | 36314103 | rs110299924 | G | A | HERC6 | HOL | 0.012 | −7.0 | −10.7 |
| 14 | 64187085 | rs210809388 | A | G | SPAG1 | HOL | 0.607 | −8.9 | N.S. |
| 6 | 85468855 | rs109381664 | G | A | HSTN | HOL | 0.353 | N.S. | −9.0 |
| 11 | 103405870 | rs133821799 | G | T | CAMSAP1 | HOL | 0.708 | −5.9 | −12.3 |
| 14 | 1107890 | rs458188308 | C | T | PYCR3 | HOL | 0.075 | N.S. | −11.1 |
| 20 | 32759553 | rs208833611 | C | T | OXCT1 | HOL | 0.693 | N.S. | −11.5 |
| 6 | 85534747 | rs110808655 | T | C | CSN1S2 | HOL | 0.360 | −12.3 | N.S. |
| 14 | 3033158 | rs208093012 | T | G | PTK2 | MON | 0.024 | N.S. | −13.5 |
| 14 | 1044777 | rs110323635 | G | A | MAPK15 | HOL / MON | 0.538 / 0.270 | −15.1 | −11.8 |
| 18 | 44773142 | rs41881032 | G | A | GARRE1 | HOL | 0.829 | N.S. | −15.2 |
| 14 | 1042664 | rs473875358 | G | A | MAPK15 | NOR | 0.034 | −16.2 | −15.0 |
| 6 | 36313547 | rs110574592 | G | T | HERC6 | HOL | 0.011 | −24.6 | −13.1 |
| 25 | 1684457 | rs462748617 | G | T | RAB26 | CHA | 0.079 | −18.9 | −20.1 |
| 19 | 61806013 | rs383537478 | C | T | ARSG | NOR | 0.182 | N.S. | −20.2 |
| 15 | 77368805 | rs136792188 | C | T | PTPMT1 | CHA | 0.405 | N.A. | −25.4 |
| 14 | 1042682 | rs109948623 | A | G | MAPK15 | HOL | 0.524 | −48.5 | −21.5 |
| 14 | 1007601 | rs110502044 | G | A | IQANK1 | HOL | 0.273 | −34.9 | −43.9 |
| 19 | 40038954 | rs133000623 | T | C | STARD3 | HOL | 0.133 | −41.3 | −39.6 |

Only significant ΔPSI values are indicated (FDR <0.01).

*NS* not significant, *NA* not available, *H* HEK293T, *M* MAC-T.

[a]Corresponds to the breed in which an association between the SDV and one or more phenotypes was measured.

[b]Corresponds to the frequency of the alternative allele in this breed.

So, the role of *DGAT1* in modifying milk phenotypes has been well characterised, and it is expected that increasing DGAT1 function increases PC and FC phenotypes, and decreases MY phenotypes[95]. On the other hand, it would be interesting to understand in future studies to what extent rs134725785 contributes to the observed effect of the pK232A-containing haplotype, or in an individual manner, on dairy traits. Finally, the rs134725785 polymorphism is located in the intron 2 of *DGAT1*, the splicing rate of which was associated with milk traits[96]. The two other identified causal variants (*PIK3C2G* rs135835897 and *PIAS4* rs133242826) were more difficult to interpret because we had to infer the role of the genes they impacted in modifying the associated phenotypes. To our knowledge, the biological role of these two genes has never been described in cattle so far. Phosphatidylinositol 3-kinase C2 domain-containing subunit gamma, encoded by the *PIK3C2G* gene,

is a lipid kinase that phosphorylates inositol phospholipids, thereby controlling membrane lipid composition and regulating a wide range of intracellular processes, including vesicular trafficking and signal transduction[97]. *PIK3C2G* knockout mice present severely reduced liver accumulation of glycogen and develop hyperlipidemia, adiposity as well and insulin resistance with age or after consumption of a high-fat diet[98]. In cattle, insulin resistance promotes the sparing of glucose, increased lipolysis in adipose tissue, and increased availability of non-esterified fatty acids for oxidation and milk fat synthesis[99]. These observations suggest that the *PIK3C2G* function modulates the FC phenotype in cattle. Protein inhibitor of activated STAT Y, encoded by the *PIAS4* gene (also known as *PIASy*), is the shortest member of the PIAS family and has been reported to modulate transcriptional activities of STAT1[100], lymphoid enhancer factor 1 (LEF-1)[101], and the

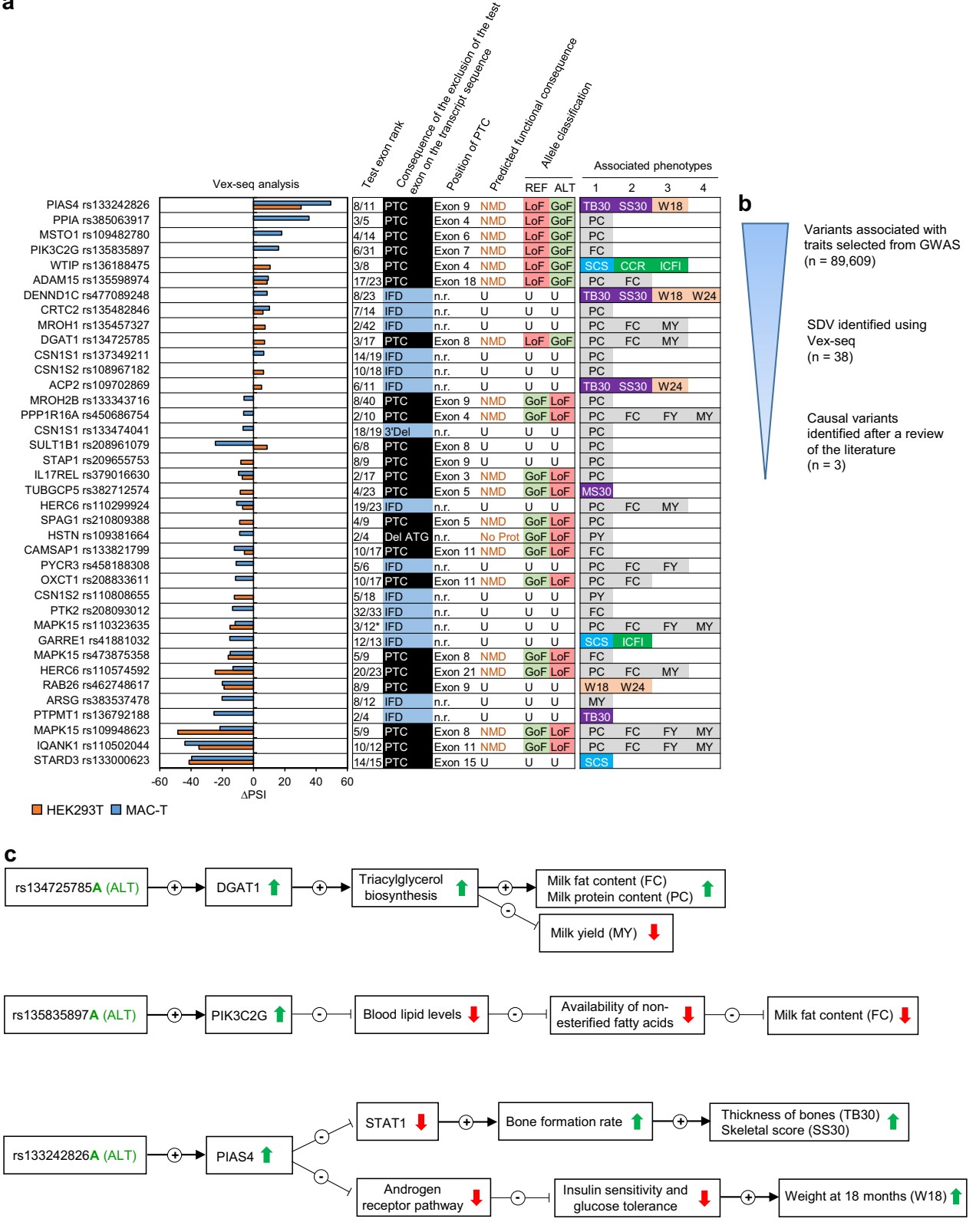

**c**

rs134725785**A** (ALT) → (+) → DGAT1 ↑ → (+) → Triacylglycerol biosynthesis ↑ → (+) → Milk fat content (FC) ↑ / Milk protein content (PC) ↑
→ (-) → Milk yield (MY) ↓

rs135835897**A** (ALT) → (+) → PIK3C2G ↑ → (-) → Blood lipid levels ↓ → (-) → Availability of non-esterified fatty acids ↓ → (-) → Milk fat content (FC) ↓

rs133242826**A** (ALT) → (+) → PIAS4 ↑
→ (-) → STAT1 ↓ → (+) → Bone formation rate ↑ → (+) → Thickness of bones (TB30) ↑ / Skeletal score (SS30) ↑
→ (-) → Androgen receptor pathway ↓ → (-) → Insulin sensitivity and glucose tolerance ↓ → (+) → Weight at 18 months (W18) ↑

androgen receptor (AR)[102,103]. PIAS4 has been characterised as a specific inhibitor of STAT1 but by a mechanism other than inhibition of STAT DNA binding, as described for PIAS1 and PIAS3[100]. The *Stat1* knockout mice had an increased bone mass[104,105], and bone morphometric analysis revealed a notable increase in bone formation rate and other osteoblast parameters such as osteoid surface/thickness and osteoblast surface. This suggests that excessive osteoblast differentiation is responsible for the increased bone mass. PIAS4 is also able to repress AR-mediated gene activation[102,103]. Progressive reduced insulin sensitivity and impaired glucose tolerance were observed in AR knockout mice with advancing age. Aging AR knockout mice displayed accelerated weight gain, hyperinsulinemia, and hyperglycaemia, and the absence of AR contributes to increased triglyceride content in skeletal muscle and liver[106]. As described above, all these observations

**Fig. 8 | Identification of splicing causal variants. a** Classification of the 38 SDV identified in the Var.GWAS dataset by functional impact type. The ΔPSI values are represented as histograms for each variant in both cell lines. The position of the test exon (rank) whose splicing is affected by the SDV is indicated in relation to all the exons of the reference transcript. The four different observed consequences on the primary structure of the transcript resulting from the exclusion of the test exon are indicated (i.e. premature termination codon (PTC), in-frame deletion (IFD), deletion in the 3′UTR (3′Del), deletion of the translation initiation codon (Del ATG)). When appropriate, the exon in which the PTC is located is indicated; nr not relevant. From this information on the transcript sequence, the consequence on protein function may be predicted. The presence of a PTC is associated with a protein loss of function via the non-sense-mediated mRNA decay (NMD) mechanism unless it is located in the last exon. The absence of the translation initiation ATG codon is associated with a protein loss of function, whereas the consequence of an in-frame deletion or a deletion in the 3′UTR is considered non-predictable; U unknown. The REF and ALT alleles of each variant whose impact on protein function has been predicted are classified as loss-of-function (LoF) or gain-of-function (GoF), depending on the ΔPSI value. Thus, an allele that increases the inclusion of the test exon required to obtain a functional protein is classified as GoF, whereas an allele that decreases the inclusion of this exon is classified as LoF. *, the *MAPK15* rs473875358 variant is located in the non-canonical ENSBTAT00000074468.1 transcript. The phenotypes associated with each variant are shown. **b** Filtering steps used to identify causal variants. **c** Illustration of the genotype-to-phenotype link for the three identified causal variants. Positive black arrows indicate that the initial condition increases the final condition in terms of function (for proteins), concentration (for compounds), or efficiency (for biological pathways or processes). Negative black arrows indicate a decrease. The resulting effect of black arrows is symbolised by a green up (increase) or red down (decrease) arrow.

are consistent with the fact that *PIAS4* is playing a role in the modulation of SS30 and TB30 phenotypes via STAT1, and in the modulation of W18 via AR. Beyond that, it should be remembered that the remaining 35 SDV represent very promising candidate causal variants, but a review of the literature and databases has not made it possible to firmly link the function of the genes they impact to the associated phenotypes, or it was not possible to conclude on their effect on protein function.

In conclusion, we have brought substantial original data to better describe SDV in cattle and understand their role in the construction of bovine phenotypes. Our study led to the identification of three putative causal variants. In addition, the tools we have adapted for cattle could be used more widely in the future for the functional annotation of genetic variants in farm animal species[6]. In particular, Pangolin and SpliceAI algorithms, which are simpler to set up than the Vex-seq method, could be used to generate high-throughput annotations of SDV for these species and, therefore, could potentially improve genomic prediction.

## Methods

### Ethics statement
All analyses were performed using data from routine recording and genotyping of French cattle in commercial herds. We did not perform any experiments on animals, and no ethical approval was required.

### Animals and phenotypes
We analysed steers, cows, or bulls from four different breeds (Holstein, Montbéliarde, Normande and Charolaise) for which phenotypes and genotypes were available. Depending on the breed, phenotypes were obtained for milk production, milk composition, fertility, mastitis resistance, growth, morphology or carcass traits. The number of animals per population ranged from 2255 to 10,066 (Table 1).

Ten traits were measured in Holstein, Montbéliarde and Normande bulls. Five traits were related to milk production and composition: milk yield (MY), fat yield (FY), protein yield (PY), fat content (FC) and protein content (PC); three female reproductive traits were also measured: the interval between calving and the first artificial insemination (ICFI) which reflects resumption of cyclicity, and heifers' (HCR) and lactating cows' (CCR) conception rates, which represent the success/failure of each artificial insemination; and two traits were related to udder health: average somatic cell score (SCS) during the whole lactation, computed as the mean of monthly records of log-transformed somatic cell counts (SCC) defined as $SCS = 3 + log2(SCC/100,000)$, and clinical mastitis (MAST) defined as at least one episode of clinical mastitis in the interval from 10 days before calving to 150 days after calving. Montbéliarde and Normande bulls were also phenotyped for live morphology traits: muscularity of the thighs (THIGHS) in both breeds and muscularity of the withers (WITHER) in Normande. Four traits were measured in Charolaise cows. Two traits were related to growth: weight at month 18 (W18) and at

month 24 (W24); and three morphology traits were measured on the living cows: muscularity score at month 30 (MS30), skeletal score at month 30 (SS30), and thickness of bones at month 30 (TB30). Three additional carcass traits were available in Montbéliarde, Normande, and Charolaise steers: age at slaughter (AS), carcass weight (CW) and carcass grade (CG).

For cow and steer populations, the traits were expressed as yield deviations (YD), i.e. mean performance adjusted for environmental effects, while daughter yield deviations (DYD) were calculated for bulls, i.e. mean performance of the daughters adjusted for environmental effects and the breeding value of their dam[107] (Table 1). For more details on the traits analysed, see our previous studies[108,109].

### Genotyping and imputation
Animals in the four breeds were genotyped with different versions of the EuroG10k (versions 1 to 7) or the 50k SNP Beadchip (versions 1 to 4), with the most recent being the Illumina EuroGMD Beadchip, which is currently used for genomic selection (https://www.eurogenomics.com/actualites/eurogenomics-new-eurog-md-beadchip.html). The standard EuroGMD Beadchip contains 53,469 autosomal SNPs that passed all quality control filters (individual call rate >95%; SNP call rate >90%; minor allele frequency (MAF) >1%; genotype frequencies in Hardy–Weinberg equilibrium with $P > 10^{-4}$). The ARS-UCD1.2 bovine genome sequence was used as a reference[110]. All imputation analyses were performed within-breed. Missing genotypes of EuroGMD SNPs are routinely imputed in the French evaluation system using FImpute software[111] by INRAE; CTIG (France, Jouy-en-Josas). Imputation to sequence level was done using a two-step approach: (1) 777k (high-density, HD) genotypes were imputed from EuroGMD genotypes using version 3 of FImpute[111], with animals with HD genotypes as a reference in each breed (776 Holstein, 522 Montbéliarde, 543 Normande and 672 Charolaise)[112] and (2) raw variants were filtered as previously described by Boussaha et al. [113] to produce 25,050,323 sequence variants (SNPs and InDels). Briefly, quality filtering was applied to the short reads aligned on the ARS-UCD1.2 reference sequence and small genomic variations (SNPs and InDels) were detected using SAMtools (v0.0.18)[114]. The selected variants were imputed with version 4 of Minimac[115] using a multi-breed population of 2255 animals from the RUN8 reference panel of the 1000 Bull Genomes consortium[1,2] and from INRAE, including 1059 Holstein, 63 Montbéliarde, 45 Normande and 147 Charolaise (Supplementary Table 2). Only variants with MAF >0.005 and an imputation $r^2$ value (estimated by Minimac) ≥0.2 were kept for GWAS.

### GWAS
The sequence variants were tested for association with different traits in each breed in separate analyses using GCTA (v1.94.1) software[116], accounting for a polygenic effect estimated with a genomic relationship matrix (GRM) from EuroGMD autosomal SNPs. The following linear mixed model was applied:

$\mathbf{y} = \mathbf{1}\mu + \mathbf{x}b + \mathbf{g} + \mathbf{e}$, where $\mathbf{y}$ is the vector of phenotypes (YD or DYD); $\mu$ is the overall mean; $b$ is the additive fixed effect of the variant tested; $\mathbf{x}$ is the vector of imputed allele dosages; $\mathbf{g} \sim N(\mathbf{0}, \mathbf{G}\,\sigma^2_g)$ is the vector of random polygenic effects, with $\mathbf{G}$ the genomic relationship matrix based on 50k SNPs, and $\sigma^2_g$ the polygenic variance; and $\mathbf{e} \sim N(\mathbf{0}, \mathbf{D}\,\sigma^2_e)$ is the vector of random residual effects, with $\sigma^2_e$ the residual variance. $\mathbf{D}$ was the identity matrix for YD analyses and a diagonal matrix with inverse weights for DYD to account for heterogeneous accuracy.

Candidate variants were selected from GWAS results based on a $-\log_{10}(p\text{ value})$ threshold $\geq 6$, where the $p$ value represents the probability associated with the effect of the tested variant. This threshold, corresponding to a 5% genome-wide threshold of significance after Bonferroni correction for 50,000 independent tests, was intentionally set relatively low to ensure a broad selection of candidate variants within the splicing regions. Confidence intervals (CI) of the QTL were defined by including variants in the upper third of the peak, extending to ±2 Mb around the lead variant, which is defined as the variant with the most significant effect.

### Generation of human and cattle sequence logos at splice sites

All available sequences at 3′ss and 5′ss regions from position −10 to +10 related to each splice site were extracted from *Homo sapiens* GRCh38 and *Bos taurus* ARS-UCD1.2 assemblies using the UCSC Table Browser (https://genome.ucsc.edu/cgi-bin/hgTables)[117]. The sequences relative to the 3′ss on one side and the 5′ss on the other side were aligned to create human and bovine frequency sequence logos. This allowed us to visualise the probability of observing each nucleotide at each position between both species. Logos were created with Seq2Logo (https://services.healthtech.dtu.dk/service.php?Seq2Logo-2.0)[118].

### Running of SpliceAI and Pangolin on the bovine genome and validation using bona fide SDV

Annotation files containing information about bovine genes, transcripts, and exons from Ensembl release 102 were specifically created in accordance with the author's recommendations to run SpliceAI (v1.3.1) and Pangolin (v1.0.2) on *Bos taurus* genome assembly ARS-UCD1.2[60,61,119]. For SpliceAI, we used a custom perl script to convert the *Bos taurus* Ensembl r102 gtf file into a GENCODE-like formatted file. For Pangolin, we used the python script create_db.py (provided in the Pangolin package) to transform a modified *Bos taurus* Ensembl r102 gtf file (with an added Ensembl_canonical tag) into the Pangolin annotation database. These custom file and database containing bovine genomic information were used instead of the usual file containing human genomic information to run SpliceAI and Pangolin with default settings on 24 bona fide bovine SDV described in the literature to assess the false negative prediction rate for both programs (Supplementary Data 1). Files generated to run SpliceAI and Pangolin are available in the Recherche Data Gouv database [https://doi.org/10.57745/UO9T9O].

### Vex-seq principles

The general principles of Vex-seq were described first in 2018[57], and have been upgraded by its authors[66]. We followed the last version of the protocol in which we made some minor modifications as specified in the further methodological section. An overview of our Vex-seq analysis is depicted in Supplementary Fig. 3.

### Selection of variants to be analysed by Vex-seq

We decided to perform a Vex-seq analysis of bovine SDV focused on different categories of variants of interest, which were divided into five specific datasets: (i) The positive control SDV includes bona fide human and bovine SDV; (ii) The Var.GWAS dataset includes variants from bovine GWAS; (iii) The Var.P dataset includes random variants intentionally enriched in putative bovine SDV using SpliceAI; (iv) The

Var.DGAT1 dataset includes variants of bovine *DGAT1*; (v) The Var.sQTL dataset includes variants from bovine sQTL studies. In view of the huge quantity and diversity of the data generated, we thought it was not suitable to present all the results obtained in a single publication. Consequently, datasets (i) to (iii) have been addressed in this study, while datasets (iv) and (v) will be addressed in subsequent publications. It should be noted that the Vex-seq execution and quality control steps are applicable to all datasets, provided that all variants were part of the same oligonucleotide pool.

Regarding datasets (i–iii), the criteria for variant selection were as follows. Positive control SDV were selected through a review of the literature and the OMIA database (https://www.omia.org/home/)[65]. SNP from GWAS located in Vex-seq test region (as defined in Supplementary Fig. 3a and next section) with an associated rsID, an imputation score ($r^2$, estimated by Minimac (v4)) above 0.4, and a $-\log_{10}(p\text{ value}) > 6$ were selected to construct the Var.GWAS dataset. Thousand random bovine variants fitting Vex-seq constraints were analysed by SpliceAI, and then a balanced selection of positive and negative predicted SDV was used to construct the Var.P dataset.

### Design of the oligonucleotide pool

As explained by ref. 57, the oligonucleotide pool was designed to include an invariable 5′ sequence [5′-CTGACTCTCTCTGCCTC-3′] directly followed by the specific test sequence carrying a unique BC, then an invariable sequence harbouring *Mfe*I and *Spe*I sites [5′-CAATTGACTACTAGT-3′], and a final invariable 3′ sequence [5′-TCTAGAGGGCCCGTTTA-3′]. In our assay, the test sequence constituted a test exon of a variable size (13 to 98 nts), systematically flanked by 50 nts of the upstream intron and 20 nts of the downstream intron. BCs were generated using the R package DNABarcodes provided by ref. 57, but 4 BCs were associated with each test sequence instead of 3. Genomic position, alleles and rsID related to variants that did not originate from GWAS were retrieved from Ensembl (release 102).

### Production and QC of the Vex-seq plasmid libraries

The plasmid library was produced through two successive steps, yielding an intermediate plasmid library (PL1) and a final plasmid library used for transfection (PL2) (Supplementary Fig. 3b, c). The modified pcAT7-Glo1 vector was a gift from ref. 57 (Addgene plasmid # 160996). Of note, our Vex-seq assay was designed to produce a PL2 made of 7352 unique constructs. The study reported here concerns only a part of them ($n = 3128$; related to datasets (i–iii) as described above; constituted from oligonucleotides described in Supplementary Data 7), as the remaining others ($n = 4224$; related to datasets (iv) and (v) as described above) will be analysed in future publications. A 10 pmole pool of 7352 oligonucleotides was produced using array-based DNA synthesis (Agilent Technologies) in order to assemble PL1 and PL2. Briefly, the modified pcAT7-Glo1 was linearised by digestion using *Pst*I and *Xba*I (New England Biolabs), the ends were then blunted using DNA Polymerase I, Large (Klenow) Fragment (Ozyme). The synthesised oligonucleotides were then PCR-amplified for 13 cycles using the KAPA HiFi Hotstart Readymix with primer pair Oligo-F/Oligo-R (Supplementary Table 3), gel purified using the QIAquick Gel Extraction Kit (Qiagen) and inserted into the linearised plasmid by means of the NEBuilder HIFI DNA Assembly Master mix (New England Biolabs). Next, the resulting plasmid pool (PL1) was digested with *Spe*I and *Mfe*I (New England Biolabs), and Exon 3 and intron 2 were PCR-amplified for 35 cycles from the original plasmid using the KAPA HiFi Hotstart Readymix with primer pair Exon 3-MfeI-F/Exon 3-XbaI-R (Supplementary Table 3). The resulting products were digested with *Mfe*I and *Xba*I and subcloned into the PL1 to obtain the final PL2.

Vex-seq plasmid libraries were sequenced to filter out non-interpretable constructs. Briefly, two PCR reactions were performed to produce two specific PL1 and PL2 libraries using the Phusion Polymerase (New England Biolabs) with 20 ng of each plasmid library, and

primer pairs PL1-F/Plasmid-R (for PL1) and PL2-F/Plasmid-R (for PL2) (Supplementary Table 3), in a final volume of 50 μL. The PCR programme had an initial denaturation at 98 °C for 30 s, 13 cycles of denaturation at 98 °C for 10 s, annealing at 65 °C for 30 s (63 °C for PL2), extension at 72 °C for 30 s, and a final extension step at 72 °C for 5 min. These PCR spanned the test region in PL1 and the exon 3 region in PL2, respectively. The primers carried a 3′ sequence targeting the plasmid construct and a 5′ sequence matching Illumina's adaptors (Illumina). To obtain the final PL1 and PL2 sequencing libraries, 10 μL of each of the above intermediate PL1 and PL2 libraries were then PCR-amplified and multiplexed using the Phusion Polymerase with Illumina's index adaptor pairs i5-UDI0001-F/i7-UDI0001-R and i5-UDI0002-F/i7-UDI0002-R (Integrated DNA Technologies) respectively (Supplementary Table 3), in a final volume of 50 μL. The PCR programme had an initial denaturation at 98 °C for 30 s, ten cycles of denaturation at 98 °C for 10 s, annealing/extension at 72 °C for 60 s, and a final extension step at 72 °C for 5 min. The length of these sequencing libraries was assessed using an Advanced Analytical Fragment Analyser (Agilent Technologies). Finally, libraries were quantified by qPCR using the Kapa Library Quantification Kit (Roche) and analysed on a MiSeq platform (Illumina) using the MiSeq Reagent Kit v2 Micro (Illumina). Read 1 and read 2 were 150 bases each. In order to ensure that BC are associated with the correct test region, BC sequences were checked using a custom Perl script, and the alignment of both reads on the test region or to exon 3 was done using bwa (v0.7.17)[120]. The QC process followed to assess PL1 and PL2 is illustrated in Supplementary Fig. 4a. In brief, a series of filters were applied to eliminate non-interpretable BC due to a failure in the synthesis or cloning of the sequences associated with them. In practical terms, BC absent in PL1 or containing synthesis errors were disregarded, as well as BC associated with a low percentage (<85%) of correct reads in PL1. BC without reads detected in PL2 were also filtered out.

## Cell culture and transfection

The HEK293T cells were provided by Sophie Dhorne-Pollet at INRAE; GABI unit (France, Jouy-en-Josas)[121]. The MAC-T cells were provided by Kathrin Kober-Rychli at the University of Veterinary Medicine; Institute of Food Safety, Food Technology and Veterinary Public Health (Austria, Vienna)[122]. HEK293T cells were cultured in the Dulbecco's modified Eagle's medium (DMEM) (Thermo Fisher Scientific) with 10% foetal calf serum (Sigma Aldrich). MAC-T cells were cultured in DMEM with 10% foetal calf serum supplemented with 4 mM L-glutamine (Thermo Fisher Scientific), 1% penicillin/streptomycin (Thermo Fisher Scientific), 1 μg/L hydrocortisone (Sigma Aldrich) and 50 mg/L insulin (Sigma Aldrich). Cells were seeded in six-well plates at a concentration of $3 \times 10^5$ cells/per well 24 hours before transfection. In each well, transfection was achieved with 1 μg of PL2 mixed with 3 μL Lipofectamine 2000 Reagent (Thermo Fisher Scientific) for HEK293T, or 2 μg of PL2 mixed with 6 μL Lipofectamine 2000 Reagent for MAC-T. The amount of transfected plasmid DNA was higher in MAC-T than in HEK293T cells due to lower transfection efficiency. Each cell line was transfected in triplicate.

## RNA extraction and reverse transcription

Total RNA was extracted from cultured cells 48 h after transfection using the RNeasy Mini Kit (Qiagen). Reverse transcription was performed using SuperScript III Reverse Transcriptase (Thermo Fisher Scientific) with the latest version of the universal molecular index (UMI) primer used by ref. 66 (5′-GTGACTGGAGTTCAGACGTGT GCTCTTCCGATCTNNNNNNNNNNGCTGATCAGCGGGTTTAAACG-3′) and following the manufacturer's guidelines. The reaction mix contained 3 μg of HEK293T total RNA or 4.5 μg MAC-T total RNA, 200 nM UMI primer, 1 mM dNTPs, 2.5 mM MgCl₂, 10 mM dithiothreitol, 40U RNAse OUT and 200U SuperScript III reverse transcriptase. The

obtained complementary DNAs were treated with 2U RNAseH (Thermo Fisher Scientific) to degrade the remaining RNA.

## Quantification of cDNA and sequencing

cDNA was PCR-amplified and multiplexed using homemade forward index primers and reverse primers corresponding to Illumina's index adaptors (Integrated DNA Technologies). Briefly, 5 μL of the cDNA obtained with each transfection triplicate in each cell line was PCR-amplified using the Phusion Polymerase with primer pairs VS-F1/i7-UDI0001-R, VS-F2/i7-UDI0002-R, VS-F3/i7-UDI0003-R, VS-F1/i7-UDI0004-R, VS-F2/i7-UDI0005-R and VS-F3/i7-UDI0006-R (Supplementary Table 3) in a final volume of 50 μL. The PCR programme had an initial denaturation at 98 °C for 30 s, ten cycles of denaturation at 98 °C for 10 s, annealing at 68 °C for 30 s, extension at 72 °C for 30 s, followed by five cycles of denaturation at 98 °C for 10 s, annealing/extension at 72 °C for 60 s, and a final extension step at 72 °C for 5 min. The obtained libraries were purified using 0.8X Ampure XP beads (Beckman Coulter), and then the quality was assessed using an Advanced Analytical Fragment Analyser (Agilent Technologies). Finally, libraries were quantified by qPCR using the Kapa Library Quantification Kit (Roche) and analysed on a MiSeq platform (Illumina) using the MiSeq Reagent Kit v3 (Illumina). Read 1 was 75 bases, and read 2 was 225 bases to ensure reading of the first splice junction (exon 1/test exon) by read 1 and reading of the UMI, BC, and the second splice junction (test exon/exon 3) by read 2.

## Analysis of Vex-seq transcripts

Reads were identified by BC, and duplicate reads were identified using the UMI and subsequently removed. For a given BC, associated reads were aligned to two reference test sequences at exon boundaries: the first one corresponding to the expected transcript carrying the test exon, the second one corresponding to the expected transcript without the test exon. Thus, reads corresponding to the transcript with or without test exon were counted in order to calculate PSI for each BC construct. Mean PSI values of all four BCs constructs corresponding to a specific variant allele, when available, were then used to calculate PSI (%) and ΔPSI (%) for each variant (Supplementary Fig. 3b). The process to ensure the reliability of the transcripts analysis is illustrated in Supplementary Fig. 4c. A series of filters were applied to eliminate non-interpretable variants. BC with less than 10 reads detected in RNA-seq were disregarded. Only variants with at least two expressed BC for each allele in each transfection experiment and a PSI >0 for at least the REF or the ALT allele were considered for the calculation of ΔPSI. The p value of ΔPSI was calculated by means of a two-tailed student's t-test, then the FDR of ΔPSI was calculated for the entire PL2 by means of the FDR online calculator (https://www.sdmproject.com/utilities/?show=FDR).

## Annotations for variant consequence and phylogenetic conservation

GERP scores and variant consequence annotations were downloaded from Ensembl (release 102).

## Prediction of mRNA transcript and protein sequences

Exon sequences were downloaded from Ensembl to predict transcript sequences with or without test exon[119]. The primary structure of proteins resulting from these transcripts was predicted using ExPASy (https://web.expasy.org/translate/)[123]. Information about PIAS4, PIK3C2G, and DGAT1 protein domains was obtained from UniProt (https://www.uniprot.org)[124]. Ensembl ID and UniProt ID of the 38 SDV from Var.GWAS are listed in Supplementary Data 8.

## Comparison between SDV from Var.GWAS and eQTL/sQTL SNP

Studies reporting both eQTL and sQTL data in cattle were collected from PubMed. SNP shared between these studies and the 38 SDV from Var.GWAS were found using the rsID.

## Reporting summary

Further information on research design is available in the Nature Portfolio Reporting Summary linked to this article.

## Data availability

The authors confirm that the summary of GWAS results, the results of SpliceAI and Pangolin predictions, in addition to the analysed data of the Vex-seq analysis, are available within the article. The GWAS data generated in this study have been deposited in the Recherche Data Gouv database [https://doi.org/10.57745/UO9T9O]. The corresponding raw phenotypic and genotypic data were produced for the purpose of bovine selection and belong to French farmers' organisations, which have given INRAE permission to use them for research purposes, excluding any transfer to third parties or public databases. They cannot, therefore, be made available to the public. Readers can request a research licence from Valogene (France, Paris) for the genotyping data and from France Génétique Elevage (France, Paris) for the phenotypic data. All MiSeq fastq files relative to the Vex-seq analysis and generated in this study have been deposited in the European Nucleotide Archive database under the accession code PRJEB87659. All files generated to run SpliceAI and Pangolin on bovine variants have been deposited in the Recherche Data Gouv database [https://doi.org/10.57745/UO9T9O]. eQTL and sQTL data from ref. 7 are available at the cattle Genotype-Tissue Expression atlas [https://cgtex.roslin.ed.ac.uk/]. eQTL and sQTL data from ref. 25 are available on Figshare [https://figshare.unimelb.edu.au/articles/dataset/eQTL_and_sQTL_from_16_cattle_tissues_linear_mixed_model_/19793047?file=35165539]. Source data are provided with this paper.

## Code availability

All custom scripts generated in this study have been deposited in the Recherche Data Gouv database [https://doi.org/10.57745/UO9T9O].

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

## Acknowledgements

The pcAT7-Glo1 vector was a kind gift from Scott I. Adamson and Brenton R. Graveley at the University of Connecticut. We thank Scott I. Adamson for his valuable advice on implementing the Vex-seq method. We thank Andrea Rau for their helpful suggestions for generating bar charts. We are grateful to the genotoul bioinformatics platform Toulouse Occitanie (Bioinfo Genotoul [https://doi.org/10.15454/1.5572369328961167E12]) for providing computing and storage resources. This work was performed in collaboration with the GeT core facility, Toulouse, France (GeT [https://doi.org/10.15454/1.5572370921303193E12]), and was supported by France Génomique National infrastructure, funded as part of 'Investissement d'avenir' programme managed by Agence Nationale pour la Recherche (contract ANR-10-INBS-09). This study was funded by the INRAE Animal Genetics division (A.B.).

## Author contributions

D.R. and A.B.: conceived and designed this study. A.B.: managed this study. M.C.: performed splicing predictions and cleaned NGS data. N.G.: performed laboratory experiments. M.B. and C.H.: managed sequence data and developed the original imputation procedure. M.-P.S. and D.B.: performed imputation and GWAS. M.C., N.G., M.-P.S. and A.B.: analysed the data. M.P.S. and A.B.: writing—original draft preparation. M.C., N.G., M.-P.S., D.R. and A.B.: writing—review and editing. All authors have read and agreed to the published version of the manuscript.

## Competing interests

The authors declare no competing interests.
