## [Transparent Peer Review file · Nature Communications]

Functional impact of splicing variants in the elaboration of complex traits in cattle

Corresponding Author: Dr Arnaud Boulling

Version 0:

Reviewer comments:

Reviewer #1

(Remarks to the Author)

Major Comments:

I think this is an interesting experiment and has been conducted with scientific rigour. I have no criticisms of the methods. However, currently the writing of the materials and methods as well as the results is less than I would expect of Nature Communications. These could be improved with a thorough review of the English and grammar and conforming to scientific writing convention. Though I note that the discussion is very well written and largely provided clarity over the results and methods that I found confusing. Things to consider:

- Lots of short paragraphs that should be combined. Often there are paragraphs that contain only one sentence, in particular in the Materials and Methods section.
- Manufacturer of all consumables and instruments should be annotated.
- The materials and methods do not provide enough detail for someone to repeat the experiment, nor do they make it clear what each step is doing and how each step is linked. See tracked comments in manuscript.
- Captions should describe the table or figure not necessarily detail the results or discuss them. Should also define axis as well as any colours used. See tracked comments in manuscript

The contribution of these methods and results to the livestock community would be very valuable. However, the writing should be improved to not only meet the journal standard but to do justice to the work being presented. I would have no hesitation in accepting the manuscript should this be addressed.

Detailed comments:

See tracked comments in manuscript

Reviewer #2

(Remarks to the Author)

In this study, authors identified 38 splice-disrupting variants (SDVs) that were associated with complex traits in cattle by integrating the significant SNPs of large-scale population-based GWAS, prediction results with SpliceAI and Pangolin, and vex-seq experiment results, of note, 3 SDVs out of these could be causal mutations. This study provided valuable new information about alternative splicing (AS) for the research community of bovine genetics and genomics, and the manuscript was well organized.

Main concern:

1. The manuscript paid more attention on how to identify SDVs associated with complex traits in bovine by using combined approach. However, the mechanism underlying alternative splicing (AS) is it impacts the protein structure or gene expression thereby impacting complex traits. Hence, I suggest more descriptions about AS would be provided in the Introduction Section, such as how many SDVs have been found until now, how AS regulates gene function, is AS tissue-specific? is the expression level of different transcripts tissue-specific?

2. Authors compared the Vex-seq results of HEK293T and Mac-T cell lines. HEK293T cells are isolated from human embryonic kidneys (HEK), and Mac-T is bovine mammary epithelial cells, I am not sure whether they are representative enough for this study. I think some ASs could be tissue-specifically occurred.

3.Regarding the 3 casual SDVs in DGAT1, PIK3C2G and PIAS4 genes, it was better to validate their expression levels in specific tissues such mammary gland and muscle etc, by using RT-PCR and qPCR thereby confirm whether the AS indeed occurred and which transcript was dominant one in specific tissues. Also, more descripts should be provided what is the differences between the transcripts of AS, such as mRNA and protein structures, whether the AS impacts the main functional region of the protein? whether and how the AS regulates the gene expression?...

Reviewer #3

(Remarks to the Author)

Charles et al 2024 performed a comprehensive analysis of cattle splicing variants. Their work covered computational analyses and wet-lab experiments and identified a set of splice-disrupting variants which were followed by further annotations and/or validations. I found their work very novel and have set up an impressive example of combining computational analyses with wet-lab technologies to identify causal variants for the field. Therefore, I think their work has a very broad impact and is certainly worth being considered to be published in this journal. The manuscript is well written. I think the majority of the work is solid, although I do have several suggestions as follows that I hope the authors can consider to improve their manuscript.

It would be good to have a main table in the main text to show the details of those 38 SDVs, including rsID, chromosome, position, alleles, frequency and etc...

While I agree with the authors that some of those 38 SDVs are missed from the single-tissue sQTL analysis of modification of splicing rate in the cGTEx polite study (as in ref 6), I can find some overlaps between 38 SDVs and sQTL identified using intron splicing ratio from (ref 15, [https://www.cell.com/cell-genomics/fulltext/S2666-979X\(23\)00182-9](https://www.cell.com/cell-genomics/fulltext/S2666-979X(23)00182-9)) in the published data of

https://figshare.unimelb.edu.au/articles/dataset/eQTL_and_sQTL_from_16_cattle_tissues_linear_mixed_model_/19793047, such as blood, milk_cell and mammary. Also, I can find 31 of 38 SDVs are significant multi-tissue sQTL from [https://www.cell.com/cell-genomics/fulltext/S2666-979X\(23\)00182-9](https://www.cell.com/cell-genomics/fulltext/S2666-979X(23)00182-9) (in the folder of "MultiTissue.sQTL" from https://figshare.unimelb.edu.au/articles/dataset/eQTL_and_sQTL_from_16_cattle_tissues_linear_mixed_model_/19793047). I have listed those 31 identified multi-tissue sQTLs that are also SDVs below:

```
SNP smt.x2 smt.x2.p smtsq.x2 smtsq.x2.p
Chr6:83318905 0.542774226491673 0.461285265736643 254.899414725888 7.38656505464567e-26
Chr6:85199301 17.512042608103 2.85493617009921e-05 730.004962197001 7.13602689451379e-101
Chr6:85468855 58.714701153064 1.82274477306115e-14 75.1582244978063 1.47060375643707e-12
Chr6:85534747 58.714701153064 1.82274477306115e-14 75.1582244978063 1.47060375643707e-12
Chr2:1268402 14.6187464748533 0.000131598923466976 470.038959229041 2.85785527120729e-54
Chr3:15041705 8.64884453137295 0.00327268836985842 6894.23833144841 0
Chr3:15541059 59.4370566562152 1.26268856135547e-14 2544.33272202252 3.07856867767148e-283
Chr3:16482608 18.7621570297528 1.48076387164144e-05 1753.81138742094 8.1611383853056e-190
Chr4:76710798 34.0394380805022 5.4006264298877e-09 664.079208632038 3.08912535679002e-76
Chr5:119386807 53.396207660038 2.72622634767491e-13 252.066542795479 5.55445878422972e-31
Chr6:83318905 0.542774226491673 0.461285265736643 254.899414725888 7.38656505464567e-26
Chr6:85199301 17.512042608103 2.85493617009921e-05 730.004962197001 7.13602689451379e-101
Chr6:85468855 58.714701153064 1.82274477306115e-14 75.1582244978063 1.47060375643707e-12
Chr6:85534747 58.714701153064 1.82274477306115e-14 75.1582244978063 1.47060375643707e-12
Chr7:17999133 0.947173909503839 0.330439699046254 838.001202389464 3.04010994650235e-120
Chr11:103405870 7.03469669973857 0.00799454117299548 2379.889285441 3.80266085596523e-274
Chr14:1042682 10.0578068314694 0.00151703712538834 5341.55119964462 0
Chr14:1007601 123.556314484645 1.05353332706458e-28 3463.618665186 0
Chr14:1044777 10.1414458198163 0.00144972411139641 5566.22221715014 0
Chr14:446321 51.1213219397549 8.6829995072314e-13 2429.87420929955 2.08118331368934e-271
Chr14:1042664 51.5783641431177 6.87955619412734e-13 1980.74146962441 4.24717184531986e-229
Chr14:3033158 41.8861323003855 9.67463617490246e-11 865.627012964265 5.31514384539854e-119
Chr14:64187085 16.6426784880757 4.51241467144265e-05 1294.32523823418 8.46609551155823e-175
Chr14:1107890 7.77906365684522 0.00528551855590064 460.178075784977 1.70331285813676e-63
Chr15:77122984 1.59345127927204 0.206833734530018 2368.03709174989 6.66013944098851e-285
Chr15:77368805 29.6372180390019 5.20950148889296e-08 1894.01500558045 1.06358124061286e-217
Chr18:44773142 8.33302484978174 0.00389307813558396 840.050668075943 1.10578336319392e-100
Chr18:44900183 71.7569588618467 2.4340402958288e-17 1042.34995626398 1.10345701307671e-125
Chr19:40038954 4.25018523295684 0.0392460495858313 449.943292713611 2.2963129131019e-59
Chr20:32759553 42.0323038787518 8.97779742344497e-11 1045.02260689244 1.00551363633158e-150
Chr20:33456383 157.691167773865 3.61511939411695e-36 1784.85595758747 5.85143604352187e-268
```

I, therefore, encourage authors to check carefully with published data and to include these results in their paper and provide discussions. I think a key point is that the multi-tissue analysis is more powerful in identifying causal regulatory variants. Also, analysis of intron splicing ratio as a computational approach may be more powerful than traditionally used PSI.

However, the multi-tissue sQTL is not clear which tissue these sQTLs are acting through, therefore, this is where the work from current authors can provide critical validations.

'We discovered one new causal SDV (rs134725785) increasing DGAT1 expression and associated to modification of milk phenotypes.' A previous study also reported that there is the splicing of a potential causal intron associated with milk traits (<https://www.biorxiv.org/content/10.1101/2022.07.13.499886v1.supplementary-material>, see attached table) which colocalises with rs134725785; this may be worthwhile reporting.

Version 1:

Reviewer comments:

Reviewer #1

(Remarks to the Author)

I am pleased to see all my concerns have been addressed. I have just a few very minor comments in the attached version. Well done on a great manuscript.

Reviewer #2

(Remarks to the Author)

The authors have already addressed all my concerns in an acceptable way and have revised the manuscript accordingly.

Reviewer #3

(Remarks to the Author)

The authors have addressed my comments and I found the paper acceptable.

Point-by-point response to the comments

First of all, we would like to thank the reviewers for their constructive comments, which helped to significantly improve the first version of the manuscript.

Changes made in response to the reviewer's comments are highlighted in yellow in the revised manuscript. We have also made some minor corrections and added some new explanations to improve the clarity and the quality of the manuscript (highlighted in green). Notably, an additional paragraph has been added in the main text in the method section to explain how the prediction of mRNA and protein sequences has been performed to assess functional consequences of SDV.

In addition, a new table (Table 3), supplementary tables (Tables S2, S6, S7, S8, and S9), and a supplementary figure (supplementary Figure 8) have been added in the revised article to show data supporting some of the results presented in the main text.

Finally, please note that the documents have been edited using Grammarly, a professional English language editor.

Reviewer's comments

Reviewer 1

Major Comments:

I think this is an interesting experiment and has been conducted with scientific rigour. I have no criticisms of the methods. However, currently the writing of the materials and methods as well as the results is less than I would expect of Nature Communications. These could be improved with a thorough review of the English and grammar and conforming to scientific writing convention. Though I note that the discussion is very well written and largely provided clarity over the results and methods that I found confusing. Things to consider:

- Lots of short paragraphs that should be combined. Often there are paragraphs that contain only one sentence, in particular in the Materials and Methods section.
- Manufacturer of all consumables and instruments should be annotated.
- The materials and methods do not provide enough detail for someone to repeat the experiment, nor do they make it clear what each step is doing and how each step is linked. See tracked comments in manuscript.
- Captions should describe the table or figure not necessarily detail the results or discuss them.

Should also define axis as well as any colours used. See tracked comments in manuscript

The contribution of these methods and results to the livestock community would be very valuable.

However, the writing should be improved to not only meet the journal standard but to do justice to the work being presented. I would have no hesitation in accepting the manuscript should this be addressed.

Detailed comments:

See tracked comments in manuscript

We agree with the general comments made by reviewer 1, and we are very grateful for the detailed comments tracked in the manuscript. First of all, here are some answers to the general points raised by reviewer 1, before providing a detailed response to all the points tracked in the document:

- Paragraphs consisting of a single sentence have been grouped together.
- Missing information on manufacturers has been added in the revised manuscript :
L.763: L-glutamine (Thermo Fisher Scientific)
L.764: penicillin/streptomycin (Thermo Fisher Scientific)
L.765: hydrocortisone (Sigma Aldrich)
L.765: insulin (Sigma Aldrich)
- The material and methods section has been significantly modified to improve clarity, in line with the comments tracked in the manuscript.
- The figure and table legends have been extensively rewritten to take into account reviewer 1's comments.

Detailed response to tracked comments in the main text:

L.51: "allow accurately pinpointing" -> "pinpoint". L.51, we made the correction.

L.53: “mainly” -> “has mainly”. L.53, we made the correction.

L.61: “A deeper understanding of the function of the genome has been considered to be a relevant input to identify causal variants by integrating biological information into statistical analyses” has been reworded as L.61: “A deeper understanding of the genome function has been considered to be relevant to help identify the variants underlying the phenotypes of interest in livestock”.

L.72: “borne in mind” has been reworded as L.71: “it is worth noting”.

L.72: “Nevertheless, it should be borne in mind that making predictions from indicators of functions (e.g. gene expression, chromatin accessibility, sequence conservation) is limited in comparison with specifically assessing the functional impact of variants. This only allows attributing probabilities on the role of each variant in phenotype variability but cannot discriminate functional variants from non-functional ones.”

Reviewer 1’s comment : why? what evidence do you have for this statement ? (and Reword last sentence).

It has been replaced by L.71: “Nevertheless, it is worth noting that making predictions from indicators of functions (e.g. gene expression, chromatin accessibility, sequence conservation) is limited in comparison with accurately assessing the functional impact of variants. A variant located in a functional region of the genome is more likely to have an impact on its function, leading to phenotypic consequences, but this information is not in itself proof of the functional nature of the variant. For example, a variant located in a regulatory region, such as a promoter, will be assigned a higher probability of being causal than variants located outside any functionally annotated regions. However, such a promoter variant may well have no effect on the expression of the gene under the control of this promoter, as illustrated by numerous studies (e.g., rs79134272 and rs476518210, rs1027363911).”

L.80: “splice-disrupting variants”

Reviewer 1’s comment : Define exactly what you mean by splice-disrupting.

A definition of splice-disrupting variant (SDV) has been added in the document, L.96: “Genetic alterations occurring in the DNA sequence of a gene and modifying the normal splice-processing of its precursor RNA are called splice-disrupting variants (SDV). This class of variant results in a modification of the mature RNA sequence by abnormal inclusion or exclusion of exonic or intronic regions from the precursor RNA”

L.81: “remained more elusive” -> “has remained more elusive”. L.101, we made the correction.

L.83 & L.85: “variations » -> “variation”. L.103 & L.105, we made the correction.

L.89 & L.90: “are” -> “were”. L.109 & L.110, we made the correction.

L.91: “as modifications at the level of transcription may also have consequences on splicing”.

Reviewer 1’s comment : what does this mean?

It has been replaced by, L.111: “as variants that alter the transcription rate can also have an effect on splicing mechanisms²⁶.”

L.96: « indicated »

Reviewer 1’s comment : wrong word

L.119, It has been replaced by “suitable”

L.98: “has been” -> “was”. L.121, we made the correction.

L.105: “spliceogenicity of genetic variants”

Reviewer 1’s comment : explain here what this means

The following explanation has been added, L.128: “which is their ability to impact the splicing of the gene in which they are located.”

L.123: paragraph formatting

L.147, we made the correction.

L.123: "SpliceAI and Pangolin are deep learning splicing prediction algorithms developed to analyse human genetic variants"

Reviewer 1's comment : How ?

The following explanation has been added, L.176: "SpliceAI has been constructed using as input only the genomic sequence of human pre-mRNA transcripts⁶⁰. It provides a score reflecting the probability that a given variant increases or decreases the efficiency of splice sites located in its vicinity. Pangolin was constructed using sequence and RNA splicing measurements from 4 tissues (*i.e.*, heart, liver, brain, testis) related to 4 species (*i.e.* human, rhesus, mouse, rat)⁶¹. Like SpliceAI, it provides a probability that a given variant impacts splicing."

L.154: (supplementary Fig. 2).

Reviewer 1's comment: caption requires more details

The legend of the supplementary Fig. 2 has been improved: "**Supplementary Figure 2.** Comparison of splice site sequences from human and cow. Pictograms display the frequency of each nucleotide at each position from -10 to +10 relative to the splice site in the 5' splice site (donor) and the 3' splice site (acceptor) regions in human and cow genes."

We also added some clarification in the main text, L.183: "This idea is supported by the similar observed frequency of nucleotides positioned close to the splice sites in both species, as illustrated by a sequence alignment of their splice regions (supplementary Fig. 2). Thus, an algorithm designed to predict SDV in human should also work efficiently in cattle."

L.157: "A review of the literature and the OMIA database (<https://www.omia.org/home/>) as comprehensive as possible allowed us to collect 24 bovine SDV supported by in vivo evidences which were analysed by SpliceAI and Pangolin"-> "A review of the literature and the OMIA database allowed us to collect 24 bovine SDV supported by in vivo evidence which we analysed with SpliceAI and Pangolin". L.189, we made the correction.

L.160: "To compare the performances of both programs, high recall (score ≥ 0.2), recommended (score ≥ 0.5) and high precision (score ≥ 0.8) cutt-offs were used as initially done to characterize SpliceAI"

Reviewer 1's comment: "cutt-offs" of what? A description of the scores would be useful here. "used" for what purpose?

Note that, L.192: "thresholds" was used instead of "cut-offs". We did it throughout the document to ensure consistency.

In addition, the following explanation was added, L.194: "As Pangolin and SpliceAI scores represent a probability for a variant to be a SDV, selecting a specific score threshold allows us to modulate the sensitivity and the specificity of the prediction. SpliceAI scores range from 0 to 1 and are classified into 4 splicing effect categories depending on their predicted effect on the strength of the indicated splice site, which are i) acceptor (AL) or ii) donor loss (DL), and iii) acceptor (AG) or iv) donor (DG) gain. Pangolin scores range from -1 to 1 and can be interpreted as a probability to modify splicing where negative scores signify a decrease in the strength of the indicated splice site, and positive scores signify an increase in the strength of the indicated splice site. You will note that to allow comparison between SpliceAI and Pangolin results, Pangolin scores are sometimes noted in absolute values in this manuscript, and SpliceAI scores for the AL and DL classes are noted in opposite numbers (negative values)."

L.162: "SpliceAI and Pangolin both returned a positive rate of 70.8% (17/24) using high recall cut-off."

Reviewer 1's comment: "positive rate" for what ?

The following explanation was added, L.206: "a positive rate of predicted SDV "

L.166: "(Fig.2C)"

Reviewer 1's comment: You use the work spliceogenicity in the caption but it's not mentioned here in the results anywhere.

The term spliceogenic has been integrated into the comment, L.207: "With regard to the consequences of SDV on gene sequence, variants modifying the canonical splice site (CSSV) were predicted to be spliceogenic with 100% accuracy, whereas intronic variants were predicted incorrectly in more than half the cases (Fig. 2c). Taking into consideration the complexity of the affected phenotypes, 88.2% (15/17) of the variants responsible for monogenic diseases are predicted to be spliceogenic whereas only 28.6% (2/7) of variants involved in complex traits are predicted to be spliceogenic (Fig. 2d)."

L.171: "Definition of Var.GWAS and Var.P datasets"

Reviewer 1's comment: below you use var.GWAS and var.P need to be consistent

We used only Var.GWAS and Var.P in the revised document.

L.174: "The var.GWAS dataset contained all 210 significant GWAS variants fitting the constraints imposed by the design of our Vex-seq assay"

Reviewer 1's comment: explain what these are here

This explanation has been added, L.220: "With our settings, technical limitations of the method only allowed the analysis of exons of 98 nt or less in length and their flanking introns, 50 nt upstream and 20 nt downstream. The first and last exons of genes and test sequences containing an MfeI or Spel restriction site were also excluded."

L.176: "Note that only significant GWAS variants with rsID and highest imputation accuracy were used"

Reviewer 1's comment: This detail doesn't appear to be in the mat and methods

Details have been added in the Materials & Methods section, L.714: "SNP from GWAS located in Vex-seq test region (as defined in supplementary Fig. 3a and next section) with an associated rsID, an imputation score (r^2) above 0.4, and a $-\log_{10}(\text{p-value}) > 6$ were selected to construct the Var.GWAS dataset".

L.176: "Note that only significant GWAS variants with rsID and highest imputation accuracy were used to facilitate variant interpretation"

Reviewer 1's comment: Not sure what this means

This explanation has been added, L.225: "to facilitate tracking variants through assembly, database browsing and ensure accuracy of the analysis".

L.178: "constituted"-> "constructed". L.226, we made the correction.

L.178: "The var.P dataset has been constituted from a pool of 1000 random bovine variants fitting Vex-seq constraints, subsequently analyzed by SpliceAI and filtered to keep only 146 variants with a balanced distribution of positive and negative scores."

Reviewer 1's comment: A description of the scores is required prior to this (in previous section) including what it means to be positive or negative

This explanation has been added in the previous section, L.194: "As Pangolin and SpliceAI scores represent a probability for a variant to be a SDV, selecting a specific score threshold allows us to modulate the sensitivity and the specificity of the prediction. SpliceAI scores are between 0 and 1 and classified into 4 splicing effect categories depending on their predicted effect on the strength of the indicated splice site, which are i) acceptor (AL) or ii) donor loss (DL), and iii) acceptor (AG) or iv) donor (DG) gain. Pangolin scores are between -1 and 1 and can be interpreted as a probability to modify splicing where negative scores signify a decrease in the strength of the indicated splice site, and positive scores signify an increase in the strength of the indicated splice site. You will note that to allow comparison between SpliceAI and Pangolin results, Pangolin scores are sometimes noted in absolute values in this manuscript, and SpliceAI scores for the AL and DL classes are sometimes noted in opposite numbers (negative values). This is specified in the figures and their legends."

L.181: "intentionally enriched with spliceogenic variants"

Reviewer 1's comment: putative, as they have no other evidence of being spliceogenic.

We changed it, L.229: "intentionally enriched with putative spliceogenic variants"

L.183: "Identification of causal variants"

Reviewer 1's comment: For what ?

This explanation has been added, L.231: "The identification of causal variants responsible for phenotype variation"

L.191: "consisted in the verification"->"consisted of verification". L.240, we made the correction.

L.193: "85% of correct reads"

Reviewer 1's comment: From where ?

This explanation has been added, L.242: "85% of correct reads from the MiSeq run"

L.194: "The second step of the QC consisted in the control of the data from the transcriptomic analysis, which aimed to eliminate variants associated with BC with too low expression or not validated at step 1"
Reviewer 1's comment: Remove "of the QC consisted in the control of the data from the transcriptomic analysis, which aimed to"; "too low": define too low; "step 1": What is step 1?

This explanation has been added, L.243: "In a second step, variants associated with BC with too low expression (< 10 reads) in the transcripts analysis were eliminated".

The term "not validated at step 1" referred to the previous sentence "This resulted in the validation of 94.03% of BCs". We removed it because it was unnecessary and unclear. Note that the term "transcriptomic" has been replaced by "transcripts", which is more appropriated.

L.199: "Vexseq"->"Vex-seq". L.248, we made the correction.

L.206: "($r^2 > 0.92$ within a particular cell type)"

Reviewer 1's comment: "Which one ? List both"

This explanation has been added, L.260: "A high reproducibility between BC replicates was observed within a given cell type ($r > 0.96$ for both HEK293T and MAC-T cells)"

L.206: "has"->"had". L.262, we made the correction.

Note that L.264-265, values of correlation coefficient were corrected in the revised manuscript (" $r > 0.94$ and > 0.92 for HEK293T and MAC-T cells, respectively) and also, albeit to a lesser extent, between cell lines ($r > 0.86$ "). They were incorrectly calculated in the main text as r^2 instead of r , as shown on the figures in the first version of the manuscript.

L.215: "midigene assays"

Reviewer 1's comment: explain what this is.

This explanation has been added, L.271: "These midigenes were splice vectors of varying lengths (up to 11.7 kb) covering almost the entire ABCA4 gene and transfected in HEK293T cells, which allowed to investigate the effect of SDV in relatively large sequence context."

L.216: "This additional group of variants with various functional impacts"

Reviewer 1's comment: Are these all from a single gene? Worth noting here

This explanation has been added, L.270: "Moreover, 13 SDV exclusively localized in human ABCA4 gene previously analysed by means of midigene assays were also included in the analysis. These midigenes were splice vectors of varying lengths (up to 11.7 kb) covering almost the entire ABCA4 gene and transfected in HEK293T cells, which allowed to investigate the effect of SDV in a relatively large sequence context."

L.219: There is no description of the results for this section

Description of these results was removed from the legend of the Supplementary Fig.7 and added to the main text, L278: "Vex-seq Δ PSI values were ranging from -9 to -88 and were highly correlated between cell lines (Supplementary Fig. 7a, b, c). In order to allow comparison between Vex-seq analysis outputs (PSI and Δ PSI) and midigene analysis outputs from the study by Sangermano et al. (% of abnormal ABCA4 transcripts associated with the ALT variant allele), we converted PSI values of REF and ALT alleles into a % of abnormal ABCA4 transcripts associated with ALT allele according to the following formula: $100 - ([PSI(ALT)/PSI(REF)] * 100)$. The comparison of this percentage for each variant obtained with Vex-seq against midigene showed both approaches yielded similar outcomes with a Pearson correlation coefficient of 0.7891 and 0.8968 in HEK293T and MAC-T cells, respectively (supplementary Fig. 7d). Vex-seq sensitivity and specificity were calculated using midigene data as benchmark. Sensitivity was between 95.45% and 100%, and specificity was between 50% and 75% (supplementary Fig. 7e, f). However, this result should be treated with caution as 13 samples is low to estimate these parameters."

L.224: Fig.3a

Reviewer 1's comment: I don't think this is required as a figure, certainly not in main paper.

Fig 3a has been removed.

L.224: "The consequence is that a nucleotide change in a 3'ss or 5'ss inevitably leads to a loss of function of the latter"

This has been specified, L.295: "The consequence is that a nucleotide change in a 3'ss or 5'ss inevitably leads to a loss of function of these splice sites"

L.228: "lead"->"led". L.299, we made this correction.

L.229: "is"->"was". L.300, we made the correction.

L.248: "supplementary Table 2"

Reviewer 1's comment: Define coding in SpliceAI - consequence field. In the table you state that outputs are explained in main text, I don't see them here. I think they need to be in the table even if you have them in the main text.

This explanation has been added in the legend of Supplementary Table 1, which is the first supplementary table to provide SpliceAI and Pangolin scores:

^aThe highest SpliceAI score of a variant is defined as the maximum score obtained in one of the four splicing effect categories: acceptor gain (AG), acceptor loss (AL), donor gain (DG), donor loss (DS). It is between 0 and 1 and can be interpreted as the probability that the variant modifies splicing. The position indicated after the semicolon provides information about the location where splicing changes relative to the variant position. (GitHub - Illumina/SpliceAI: A deep learning-based tool to identify splice variants). ^bThe most extreme Pangolin score of a variant is defined as the most extreme negative or positive score attributed to the variant. Pangolin scores are comprised between -1 and 1. They can be interpreted as a probability to modify splicing where negative scores signify a decrease in the splicing strength, and positive scores signify an increase in the splicing strength. The position indicated first provides information about the location where splicing changes relative to the variant position. (GitHub - tkzeng/Pangolin: Pangolin is a deep-learning method for predicting splice site strengths.) "

Legends of supplementary Tables 3 and 4 now refer to this explanation.

L.291: "Scatter plots between Δ PSI values and GERP scores showed a significant enrichment"

Reviewer 1's comment: scatter plots don't test enrichment. how was this tested?

This was tested using a Fisher's exact test. This explanation was added, L.362: "Scatter plots between Δ PSI values and GERP scores showed significant enrichment of variants affecting phylogenetically conserved nucleotides within the group of variants decreasing splicing rate (Fig. 6a, c). This was statistically confirmed by an enrichment test which exhibited a ~1.8-fold increase (two-tailed Fisher's exact test; p-value<0.0001) of conserved variants in the "Loss" category compared to "Neutral", observed in both cell lines (Fig. 6b, d)."

L.318: "From this was predicted,"

Reviewer 1's comment: poor wording

This has been replaced, L.395: "This information was used to predict"

L.322: "functional impact of the variant on the gene function;"

Reviewer 1's comment: explain in more detail. Do you just mean that you confirm it an SDV?

This explanation has been added, L.398: "(ii) a predicted functional impact of the variant on the gene function (*i.e.*, a gain or a loss of protein expression/function resulting from the splicing disruption);"

L.322: "a functional link between the impacted gene and the considered phenotype"

Reviewer 1's comment: how do you determine this?

This explanation has been added to clarify this, L.401: "The last point can be documented by direct experimental demonstration using rare natural variants with large effect size identified in cattle⁴⁵, or indirectly inferred by connecting multiple scientific evidences together to reconstitute the biological process involved in the gene-phenotype relationship."

L.323: "(Fig. 8b)"

Reviewer 1's comment: not required as a figure

Fig.8b has been removed in the revised manuscript.

L.325: "supplementary Text"

Reviewer 1's comment: this is discussion

The supplementary text has been removed and the information about the role of *DGAT1*, *PIK3C2G* and *PIAS4* have been included in the discussion, from L.536 to L.580.

L.356: "Only one variant showed an inverse effect in the two cell lines (SULT1B1 rs208961079), what is interesting to underline because the tissue-specific nature of certain SDV is still a burning question"

Reviewer 1's comment: I don't understand this sentence, reword

This has been replaced by L.464: "Further functional characterization of this SDV could lead to a better understanding of the impact of genetic alterations on the tissue-specific regulation of splicing, which remains a burning question."

L.360: "in a lesser extend" -> "to a lesser extent". L.468, we made the correction.

L.375: "they perform" -> "they would perform". L.482, we made the correction.

L.380: "This was very similar to the SpliceAI sensitivity (89.9%) calculated with the same threshold on the basis of human SDV identified in the context of the molecular diagnostic of genetic disorders"

Reviewer 1's comment: poorly worded

This has been replaced by L.487: "This result was very close to the sensitivity of SpliceAI (89.9% using the same threshold) calculated from predictions made for human SDV identified in the context of the molecular diagnosis of genetic disorders"

L.400: "which may hold potential value in a perspective of further functional validation."

Reviewer 1's comment: context rather than perspective?

L. 507, this has been simplified: "which may hold potential value for further functional validation."

L.428: "(see also Supplementary Text)".

Reviewer 1's comment: I think this is relevant to add to the main text

The supplementary text has been removed and the information about the role of *DGAT1*, *PIK3C2G* and *PIAS4* have been included in the discussion, from L.536 to L.580.

L.430: "to" -> "with". L.539, we made the correction.

L.435: "Two other new causal variants (*PIK3C2G* rs135835897 and *PIAS4* rs133242826) have been identified. To our knowledge, the biological role of these two genes have never been described in cattle so far."

Reviewer 1's comment: Supplementary text would be valuable here for these two variants

The supplementary text has been removed and the information about the role of *DGAT1*, *PIK3C2G* and *PIAS4* have been included in the discussion, from L.536 to L.580.

L.469: "average somatic cell score (SCS) at lactation level"

Reviewer 1's comment: what does this mean, number of lactations?

This explanation has been added, L.611: "average somatic cell score (SCS) during the whole lactation"

L.470: "computed as the mean of monthly records of log-transformed somatic cell counts, and clinical mastitis"

Reviewer 1's comment: A formula would help to understand this, or improve the wording

A formula has been added, L.612: "somatic cell counts (SCC) defined as $SCS = 3 + \log_2(SCC/100,000)$ "

L.484: "mean performance of the daughters adjusted for environmental effects and the breeding value of their dam"

Reviewer 1's comment: offspring? Or does this only apply to dairy traits measured in cows?

Yes, as indicated L.463-475 in the first version of the manuscript, DYD were computed for dairy traits measured on daughters of bulls.

L.489: "Animals in the four breeds were genotyped with different versions of the 50k SNP Beadchip"

Reviewer 1's comment: You only detail one, you should list all

More details on the different versions of chips were added, L.630: "Animals in the four breeds were genotyped with different versions of the EuroG10k (versions 1 to 7) or the 50k SNP Beadchip (versions 1 to 4)"

L.490: "EuroGMD Beadchip »

Reviewer 1's comment: list manufacturer

The manufacturer has been listed, L.632: "Illumina EuroGMD Beadchip"

L.495: "Genotypes were aligned on the ARS-UCD1.2 bovine reference genome sequence"

Reviewer 1's comment: We don't align genotypes to a reference. Do you mean that all variant coordinates were from ARS-UCD1.2?

The sentence was reformulated, L.636: "The ARS-UCD1.2 bovine genome sequence was used as a reference."

L. 497: "Imputation at the sequence level was done" -> "Imputation to sequence level was done". L.639, we made the correction.

L.502: "using a multi-breed population of 1,479 animals"

Reviewer 1's comment: which breeds?

L.647, we have updated the manuscript by including the number of breeds and adding a supplementary table (Supplementary Table 7) that lists all breeds along with the corresponding number of animals for each breed. Additionally, we have corrected the inaccuracies in the previously reported numbers (highlighted in green).

L.502: "1,479 animals from the RUN8 reference panel of the 1000 Bull Genomes consortium"

Reviewer 1's comment: What filters were applied to this reference panel? Or did you use all raw variants?

This explanation and a new ref (114) were added, L.643: "raw variants were filtered as previously described¹¹⁴ to produce 25,050,323 sequence variants"

L.504: "Only variants with MAF > 0.005 were kept for GWAS."

Reviewer 1's comment: Did you apply any filter to the R2 imputation scores?

Yes, we added this information, L.647: "Only variants with MAF > 0.005 and an imputation r² value (estimated by Minimac) ≥ 0.2 were kept for GWAS."

L.509: "accounting for a polygenic effect estimated from EuroGMD autosomal SNPs"->

Reviewer 1's comment: estimated as a genomics relationship matrix (GRM) calculated from

L.652, we made the correction: "with a genomic relationship matrix (GRM)"

L.529: "There were aligned from positions -10 to +10 related to each of 3'ss and 5'ss to create sequence logos for each of these species. Logos were created with Seq2Logo (<https://services.healthtech.dtu.dk/service.php?Seq2Logo-2.0>)"

Reviewer 1's comment: This step is unclear. Explain why you created logos. What is the input? What logo type, clustering method and threshold did you use?

We have added more details to explain how and why we created logos, L.672: "All available sequences at 3'ss and 5'ss regions from position -10 to +10 related to each splice site were extracted from *Homo sapiens* GRCh38 and *Bos taurus* ARS-UCD1.2 assemblies using the UCSC Table Browser (<https://genome.ucsc.edu/cgi-bin/hgTables>)¹¹⁷. The sequences relative to the 3'ss on one side and the 5'ss on the other side were aligned to create human and bovine frequency sequence logos. This allowed us to visualize the probability of observing each nucleotide at each position between both species." We did not carry out statistical tests to compare the results of the 2 species. The logos are used to illustrate the frequencies of nucleotides in each species visually."

L.533: "with"->"of". We did not make the correction but reworded it, L.680: "and validation using bona fide SDV", because SDV validated in vivo have been used to validate the prediction programs.

L.537: "SpliceAI and Pangolin were run on 24 cattle SDV described in the literature to assess false negative rate"

Reviewer 1's comment: with default settings?

Reviewer 1's comment: Did you run it on the above mentioned annotation file?

Reviewer 1's comment: "to assess false negative rate" of what?

We have added more details to explain how we used the prediction programs, L.684: "For SpliceAI, we used a custom perl script to convert the *Bos taurus* Ensembl r102 gtf file into a GENCODE-like formatted file. For Pangolin, we used the python script `create_db.py` (provided in the Pangolin package) to transform a modified *Bos taurus* Ensembl r102 gtf file (with added Ensembl_canonical tag) into the Pangolin annotation database. These custom file and database containing bovine genomic information

were used instead of the usual file containing human genomic information to run SpliceAI and Pangolin with default settings on 24 bona fide bovine SDV described in the literature to assess the false negative prediction rate for both programs.”

L. 547: “a”->“an”. L.722, we made the correction.

L.548: “specific test sequence”

Reviewer 1’s comment: There is no detail on what the test sequences were in the materials and methods L.740, the Supplementary Table 8 containing the sequences of oligonucleotides has been added.

L.555: “Variants analysed that did not originate from GWAS were retrieved from Ensembl”

Reviewer 1’s comment: what was retrieved from Ensembl?

This explanation has been added, L.730: “Genomic position, alleles and rsID related to variants that did not originate from GWAS were retrieved from Ensembl (release 102).”

L.561: “our Vex-seq assay was designed to produce a PL2 made of 7,354 unique constructs”

Reviewer 1’s comment: You have not detailed how these were selected here in the materials and methods

A new paragraph entitled “*Selection of variants to be analysed by Vex-seq*” describing the selection process of variants to be analysed by Vex-seq has been added, from L.699 to L.719.

We also gave more details about how we obtained the oligopool, L.741: “A 10 picomoles pool of 7,352 oligonucleotides was produced using array-based DNA synthesis (Agilent Technologies), and then PL1 and PL2 were assembled as previously described in the improved version of the Vex-seq method”

L.563: “The study reported here concerns only a part of them (3,128),”

Reviewer 1’s comment: What criteria have you used to make this decision?

We have given the explanation in the new paragraph *Selection of variants to be analysed by Vex-seq*, L.706: “In view of the huge quantity and heterogeneity of the data generated, we thought it was not suitable to present all the results obtained in a single publication. Consequently, datasets (i) to (iii) have been addressed in this study, while datasets (iv) and (v) will be addressed in subsequent publications. It should be noted that the Vex-seq execution and quality control steps are applicable to all datasets, provided that all variants were part of the same oligonucleotide pool.”

Also, we mentioned the exact number of constructs which were included in this study, L.738: “(n = 3,128; related to datasets (i)-(iii) as described above; constituted from oligonucleotides described in supplementary Table 8), as the remaining others (n = 4,224; related to datasets (iv) and (v) as described above) will be analysed in future publications.”

L.564: “Nevertheless, the Vex-seq QC and the calculation of the FDR were carried out on all the constructs.”

Reviewer 1’s comment: how ?

This sentence has been removed, and the information about statistics is later in the paragraph *Analysis of Vex-seq transcripts*, L.809: “The p-value of Δ PSI was calculated by means of a two-tailed student’s t-test, then the FDR of Δ PSI was calculated for the entire PL2 by means of the FDR online calculator (<https://www.sdmproject.com/utilities/?show=FDR>).”

The information about execution and QC of the Vex-seq has been added before in the manuscript, L. 709: “It should be noted that the Vex-seq execution and quality control steps are applicable to all datasets, provided that all variants were part of the same oligonucleotide pool.”

L.566: “The quality of the obtained sequencing libraries”

Reviewer 1’s comment: what aspects of quality?

We have added more details. Here, the quality of the sequencing libraries means the length of the amplicons, L.745: “Vex-seq plasmid libraries were sequenced to filter out non-interpretable constructs. PL1 and PL2 were PCR amplified and multiplexed as previously described⁵⁷. The length of the obtained sequencing libraries was assessed using an Advanced Analytical Fragment Analyser (Agilent Technologies).”

L.570: "BC sequences were checked using a custom perl script"

Reviewer 1's comment: for what ?

An explication has been added, L.750: "In order to ensure that BC are associated with the correct test region, BC sequences were checked using a custom Perl script".

L.571: "the alignment of reads on the test region or on the exon 3 was done using bwa"

Reviewer 1's comment: Does this mean you only had two contigs in your reference sequence?

Yes, we have specified that, L.752: "and the alignment of both reads on the test region or to exon 3 was done using bwa (v0.7.17)".

L.572: "The QC process followed to assess PL1 and PL2 is detailed in the results section and in supplementary Fig. 4"

Reviewer 1's comment: should be detailed in mat and meth section rather than results

This information has been added in the main text, L.754: "In brief, a series of filters were applied to eliminate non-interpretable BC due to a failure in the synthesis or cloning of the sequences associated with them. In practical terms, BC absent in PL1 or containing synthesis errors were disregarded, as well as BC associated with a low percentage (<85%) of correct reads in PL1. BC without reads detected in PL2 were also filtered out."

L.587: "Total RNA was extracted 48 hours after transfection by means of the RNeasy Mini Kit"

Reviewer 1's comment: extracted from what ?

This has been detailed, L.774: "Total RNA was extracted from cultured cells".

Reviewer 1's comment: by means of -> using. L.774, we made the correction

L.589: "with the latest version of the UMI primer"

Reviewer 1's comment: define

We define UMI, L. 776: "universal molecular index (UMI)".

L.592: "The reaction mix contained 3 µg of HEK293T total RNA or 4.5 µg MAC-T total RNA, 200 nM UMI primer, 1 mM dNTPs, 2.5 mM MgCl₂, 10 mM dithiothreitol, 40U RNase OUT and 200U SuperScript III reverse transcriptase. The obtained complementary DNAs were treated with 2U RNaseH (Thermo Fisher Scientific) to degrade the remaining RNA."

Reviewer 1's comment: Not required if according to manufacturer's guidelines

Some parameters vary from the default conditions. For this reason we preferred to give detailed indications.

L.599: "were"->"was". L.786, we made the correction.

L.599: "cDNA were PCR amplified and multiplexed as previously described".

Reviewer 1's comment: What library making kit was used?

We have added some details, L.786: "cDNA was PCR amplified and multiplexed using home-made forward primers and reverse index PCR primers (Integrated DNA Technologies) in conditions previously established by Adamson et al.⁵⁷"

L.603: "Read 1 was 75 bases and read 2 was 225 bases."

Reviewer 1's comment: Why different read lengths?

This explanation has been added, L.792: "Read 1 was 75 bases, and read 2 was 225 bases to ensure reading of the first splice junction (exon 1/test exon) by read 1 and reading of the UMI, BC, and the second splice junction (test exon/exon 3) by read 2."

L.607: "Reads were identified by BC and duplicate reads were then collapsed into a single read"

Reviewer 1's comment: how ?

We have changed the term "collapsed" to be more accurate, L. 797: "Reads were identified by BC, and duplicate reads were identified using the UMI and subsequently removed."

L.608:"on"->"to". L.798, we made the correction.

L.608:"associated reads were aligned on two reference test sequences at exon boundaries"

Reviewer 1's comment: remove "at exon boundaries"

L.798, we made the correction

L.612:"the" -> remove. L.802, we made the correction

L.614:"The QC process of the transcriptomic analysis is detailed in the results section and in supplementary Fig. 4"

Reviewer 1's comment: should be detailed in the mat and methods

This information has been added in the main text, L.804: "The process to ensure the reliability of the transcripts analysis is illustrated in supplementary Fig. 4c. A series of filters were applied to eliminate non-interpretable variants. BC with less than 10 reads detected in RNA-seq were disregarded. Only variants with at least 2 expressed BC for each allele in each transfection experiment and a PSI > 0 for at least the REF or the ALT allele were considered for the calculation of Δ PSI."

L.617, Reviewer 1's comment: The mat and methods do not detail the how relationships between GWAS and Vex-Seq are undertaken.

The method used to construct the Var.GWAS dataset has been detailed in the new paragraph *Selection of variants to be analysed by Vex-seq*, L.714: "SNP from GWAS located in Vex-seq test region (as defined in supplementary Fig. 3a and next section) with an associated rsID, an imputation score (r^2) above 0.4, and a $-\log_{10}(\text{p-value}) > 6$ were selected to construct Var.GWAS."

L.826, Reviewer 1's comment: Figure 2, Brackets in legend for D look incorrect.

We used square brackets to note intervals, by following European notation rules. We changed this to follow international rules, where [is inclusive and (is exclusive.

L827:"constituted"->"compiled". L.1142, we made the correction.

L.835:"Proportion of variants predicted to be spliceogenic or non spliceogenic classified by variant consequence"

Reviewer 1's comment: Explain the colour categories, axis etc

This explanation has been added, L.1152: "Graphs displaying stacked green and purple bars indicate SpliceAI and Pangolin predictions, respectively. The number of variants for each score category is shown on the Y-axis, with the highest categories shown in the darkest colours. Variants with a score equal to or higher than 0.2 are predicted to be SDV. Pangolin scores are represented in absolute values."

L.842:"showed"->"shown". L.1158, the legend of the Fig. 3,a) has been corrected.

L.845:"variants from complete Var.GWAS and Var.P datasets."

Reviewer 1's comment: reword

This has been replaced by, L.1163: "Average PSI for reference and alternative alleles of variants from whole Var.GWAS and Var.P datasets"

L.846:"Proportion of SDV determined by dataset"-> "proportion of tests confirmed to be SDV"

We used this formulation, L.1164: "Proportion of tested variants confirmed to be SDV determined for each dataset"

L.847:"are"->"were". L.1166, we made the correction

L.848:"The majority of SDV overlap"

Reviewer 1's comment: result not description

The legend of Fig.3,f) has been reworded to be descriptive, L.1167: "f) Overlap of SDV from merged Var.GWAS and Var.P datasets between HEK293T and MAC-T cell lines. Scatter plot of g) PSI and h) Δ PSI values from merged Var.GWAS and Var.P datasets in HEK293T versus MAC-T cells. r , Pearson correlation coefficient."

L.854: "SpliceAI and Pangolin scores are highly correlated"

Reviewer 1's comment: this is result rather than description

The legend of Fig.4,b) has been reworded to be descriptive, L.1174: "b) Scatter plot of SpliceAI versus Pangolin scores. SpliceAI scores related to donor or acceptor loss have been turned into opposite numbers to allow comparison with Pangolin scores. r , Pearson correlation coefficient."

L.857: "Pangolin scores showed better correlation with Δ PSI than SpliceAI scores"

Reviewer 1's comment: result not description

The legend of Fig.4,c) has been reworded to be descriptive, L.1176: "c) Scatter plot of SpliceAI and Pangolin scores each versus Δ PSI values in both HEK293T or MAC-T cell lines."

L.859: "Pangolin displayed better specificity and sensitivity than SpliceAI."

Reviewer 1's comment: result not description

The legend of Fig.4,e) has been reworded to be descriptive, L.1181: "e) Receiving operating curve (ROC) and area under the curve (AUC) were calculated for SpliceAI and Pangolin in each cell line and for the mean values between the two cell lines."

L.875: Reviewer 1's comment: Figure 6, ensure caption is description not statement of result

L.1189, the legend of Figure 6 has been completely rewritten.

L.886: Reviewer 1's comment: Figure 7, ensure caption is description not statement of result

L.1203, the legend of Figure 7 has been completely rewritten.

L.900: Reviewer 1's comment: Figure 8, Description completely inadequate. It's a complex figure that requires careful description of what you are presenting, defining acronyms is not enough

L.1210, the legend of Figure 8 has been completely rewritten.

L.906: Reviewer 1's comment: Figure 8d, I don't understand this figure at all. what do arrows mean? what is the relationship between each piece of text? Does the first arrow (under gene name) mean an increase in the splicing or an increase in gene expression or both?

This has been clarified in the Fig.8,d) section, L.1230: "c) Illustration of the genotype-to-phenotype link for the 3 identified causal variants. Positive black arrows indicate that the initial condition increases the final condition in terms of function (for proteins), concentration (for compounds), or efficiency (for biological pathways or processes). Negative black arrows indicate a decrease. The resulting effect of black arrows is symbolized by a green up (increase) or red down (decrease) arrow."

Detailed response to tracked comments in the Supplementary data

Supplementary Fig. 4

4,b) "Integrity" What does it mean ?

This information has been added instead of "integrity": "The absence of BC sequence alteration"

4, c) "Quality control of transcriptomic analysis" Reword

This has been turned into "Filtering process to select interpretable variants in transcripts analysis."

4, c) This is not a description of the plots, rather the results and some discussion.

Results from Supplementary Fig. 4 have been removed and added in the main text, L.249: "However, it should be noted that the percentage of test exons that could be analysed was about 84% in both cell lines. This difference can be explained by the fact that, of the 919 variants initially included in the whole Vex-seq plasmid library, 99 are located in exon 8 of the *DGAT1* gene, and the majority of them did not pass the QC because exon 8 was not spliced in most cases. These 99 variants are not related to the Var.GWAS or Var.P datasets (see materials and methods for details), but their failure to pass QC filters reduces the overall rate of variants validated for the whole Vex-seq analysis. If we focus specifically on datasets presented in this study, 87.1% (183/210) and 82.9% (121/146) of variants were validated for Var.GWAS and Var.P dataset, respectively."

4, d) "Overlapped" -> "Overlap". We made the correction.

Supplementary Fig. 5

We have replaced the two-colour system (red and blue) by a single colour (green), which makes it easier to read and understand the plot. A color scale has been added in each figure panel, and an explanation, "dark green represents higher values" has been added in the legend.

Supplementary Fig. 6

A panel has been dedicated to each variant in order to clarify the figure. Also, non-essential information has been removed (the exact values of the Δ PSI and FDR, which are now in the supplementary table S2).

Supplementary Fig. 7

The letters H and M have been removed from the histogram in panel b). Legends of panels c) and d) have been reworded to be descriptive: "c) Scatter plot of Δ PSI in HEK293T versus MAC-T cells. r, Pearson correlation coefficient. d) Scatter plot of the percentage of abnormal transcript measured using midigenes versus Vex-seq."

The forgotten captions for panels e) and f) have been added: "e) Number of true positive (TP), false positive (FP), true negative (TN), and false negative (FN) in HEK293T and MAC-T cells depending on the FDR threshold and using ABCA4 variants analysed by midigene as a benchmark. f) Calculation of the sensitivity and specificity of Vex-seq according to (e)."

Reviewer 2

In this study, authors identified 38 splice-disrupting variants (SDVs) that were associated with complex traits in cattle by integrating the significant SNPs of large-scale population-based GWAS, prediction results with SpliceAI and Pangolin, and vex-seq experiment results, of note, 3 SDVs out of these could be causal mutations. This study provided valuable new information about alternative splicing (AS) for the research community of bovine genetics and genomics, and the manuscript was well organized.

Main concern:

1. The manuscript paid more attention on how to identify SDVs associated with complex traits in bovine by using combined approach. However, the mechanism underlying alternative splicing (AS) is it impacts the protein structure or gene expression thereby impacting complex traits. Hence, I suggest more descriptions about AS would be provided in the Introduction Section, such as how many SDVs have been found until now, how AS regulates gene function, is AS tissue-specific? is the expression level of different transcripts tissue-specific?

We agree with Reviewer 2 that more information on alternative splicing should be given in the introduction. We have added a new paragraph to give basic information on AS, with several new bibliographic references (from L.82 to L.95):

"Splicing is the process by which introns are removed from the primary messenger RNA (mRNA) transcript, and the exons are joined together to obtain a mature mRNA^{12,13}. Alternative splicing refers to the process where different combinations of exons from the same gene can be joined or skipped, resulting in diverse mRNA transcripts that encode proteins with varied structures and functions. This mechanism allows for greater complexity of the proteome and participates in phenotypic diversity^{12,14}. It also controls mRNA transcript abundance through the non-sense mediated mRNA decay (NMD) and other RNA degradation mechanisms^{15,16}. Therefore, alternative splicing represents a central element in gene expression, and it often occurs in a developmental, tissue-specific, or signal transduction-dependent manner^{16,17}. Transcriptomic studies have shown that alternative splicing is prevalent across eukaryotes and, for instance, affects the expression of 90 to 95% of human genes^{18,19}. In addition, genetic mutations are an important driver of altered gene expression and may generate novel splice patterns, thus contributing to the emergence of alternative mRNA transcripts^{14,20}."

We have also specifically mentioned the SDV validated in cattle for monogenic and complex traits (from L.113 to L.116):

"Apart from that, few studies have directly documented the effect of SDV in cattle. To our knowledge, 17 SDV responsible for monogenic diseases²⁷⁻⁴³ and 7 SDV involved in complex traits⁴⁴⁻⁵⁰ in cattle have been identified."

2. Authors compared the Vex-seq results of HEK293T and Mac-T cell lines. HEK293T cells are isolated from human embryonic kidneys (HEK), and Mac-T is bovine mammary epithelial cells, I am not sure whether they are representative enough for this study. I think some ASs could be tissue-specifically occurred.

This is a legitimate question, and some might regret that we did not test other cell types or more different cell types. Actually, it is a perfectly considered choice. We have added a paragraph to the discussion to explain why (from L.448 to L.460):

“MPRA for splicing is time-consuming, costly, and technically complex, and for this reason, only one or two cell lines are usually employed at the same time^{55,57,58}. We decided to use one bovine cell line and one human cell line already used as models to study splicing in these two species. HEK293T cells have been very frequently used to perform low-throughput splicing assays (see examples in human^{83,84} and cattle^{46,85}) but also in the context of the two other MPRA methods developed to analyse SDV^{55,58}. In some cases, comparisons with *in vivo* data from patients have been done and yielded consistent results⁵⁵. HEK293T cells have also been used to generate the ABCA4 benchmark dataset we used to validate the Vex-seq⁶⁷. MAC-T cells have been widely used to model epithelial cells of the mammary gland (for a recent example, see⁸⁶). Since the majority of SNP identified in our GWAS concern milk traits, the choice of this cell line was considered to be the most appropriate. In addition, MAC-T cells have already been successfully used to develop splicing assays for bovine genes^{85,87,88}.”

Beyond that, tissue-specific SDV are not very common. As observed by Adamson *et al.* (ref. 57 in the revised manuscript), the Vex-seq results obtained in 2 different cell types for a pool of 2059 variants are for the most part very similar. This means that the choice of cell type has relatively little impact on the results, except for a few variants that can have a tissue-specific effect.

3.Regarding the 3 casual SDVs in DGAT1, PIK3C2G and PIAS4 genes, it was better to validate their expression levels in specific tissues such mammary gland and muscle etc, by using RT-PCR and qPCR thereby confirm whether the AS indeed occurred and which transcript was dominant one in specific tissues. Also, more descripts should be provided what is the differences between the transcripts of AS, such as mRNA and protein structures, whether the AS impacts the main functional region of the protein? whether and how the AS regulates the gene expression?...

In fact, RT-qPCR analysis could be used to quantify the level of expression of the different *DGAT1*, *PIK3C2G* and *PIAS4* transcripts in the tissues related to the studied phenotype and for different genotypes. Unfortunately, we don't have any such samples, which would have to be obtained. According to Nolan, Huggett and Sanchez (Good practice guide for the application of quantitative PCR (qPCR), LGC (2013), it is admitted that at least 6 biological replicates of a given *in vivo* condition are needed to perform accurate RT-qPCR (*i.e.* in our situation: REF/REF, REF/ALT or ALT/ALT for each causal variant). Especially if the differences in terms of expression are weak between conditions.

We considered two different options for collecting these samples:

The first option was to identify animals on French farms carrying the three genotypes (REF/REF, REF/ALT or ALT/ALT) to be tested for all 3 three genes, using already available genotyping data from the animals used to perform the GWAS. Muscle and mammary gland biopsies were then taken for RNA extraction and RT-qPCR. The problem was that we would have had to wait for the animals to die before collecting the tissues after slaughter, as it would have been impossible to take these samples from living animals due to regulatory constraints. Over and above the costs associated with travel and manpower to carry out the sampling, this strategy proved to be unthinkable, as it was far too long and therefore unfeasible within the timeframe of the revisions for this article.

The second option was to obtain these samples from partners, using RNA samples that had already been extracted and stored. It would have been necessary to contact the various research teams likely to have such samples, at international level, which would have been difficult. These teams need to know the genotype of their samples for the 3 variants concerned, otherwise they will have to genotype them from the corresponding DNA samples. Furthermore, the probability of identifying samples with the rarest genotype is quite low. Considering the frequency of rarest genotypes for *PIAS4* (0.045), *PIK3C2G* (0.007) and *DGAT1* (0.005) the number of samples required to have a chance to find 6 samples for the rarest genotype in the right breed may be very high: n=134 (*PIAS4*), n=857 (*PK3C2G*), n=1200 (*DGAT1*). Taking the necessary steps to obtain these samples, if they exist, was also very time-consuming, not least because of the administrative delays involved in transferring them in due form.

Faced with these insurmountable difficulties in the time available, we abandoned the idea of obtaining these samples and performing RT-qPCR. By chance, and thanks to the suggestions of reviewer 3, we identified new exploitable data from the literature concerning sQTL and eQTL analyses. These data

enabled us to in vivo validate the effect of 27 out of 38 SDV, which colocalize with eQTL/sQTL SNPs. Descriptions of these results have been added to the main text and are available in detail in the supplementary Table 6 (from L.410 to L.419):

“Colocalization between SDV from Var.GWAS and eQTL/sQTL SNP

We searched for shared SNP between the 38 SDV from Var.GWAS and eQTL/sQTL data, through a review of published studies reporting both eQTL and sQTL SNP. Four studies did not allow the identification of shared SNP^{74–77}. Two studies conducted on imputed WGS and involving large numbers of animals and samples showed positive results with respect to single-tissue eQTL/sQTL SNP^{7,25} and multi-tissue eQTL/sQTL SNP²⁵ (supplementary Table 6). SDV colocalize with 6 single-tissue eQTL SNP and 8 single-tissue sQTL SNP. Multi-tissue analysis yielded better results, indeed, SDV also colocalized with 23 multi-tissue eQTL SNP and 23 multi-tissue sQTL SNP. Globally, 71.1% of SDV (27/38) colocalize with at least one single- or multi- e/sQTL SNP.”

We also discussed these new results in the main text (from L.525 to L.532):

“Molecular QTL data from two studies showed that 27 out of the 38 SDV colocalize with *cis*-sQTL and *cis*-eQTL SNP in cattle^{7,25}. This finding supports that these variants are true SDV. It should be noted that these two studies led to the identification of colocalized SDV as they were carried out on a large number of animals and samples, allowing the identification of many molecular QTL. Also, the multi-tissue analyses are more powerful and identify more QTL than single-tissue ones, which also explains why they colocalized with a greater number of SDV. Ten SDV did not colocalize with any molecular QTL, but this can be explained in various ways. The consequences of a SDV may be undetectable by sQTL or eQTL analyses if it is not expressed in the tissues tested or at the right timing. The power and completeness of these QTL analyses is also limited by the size and genetic background of the samples available.”

Unfortunately, none of the 3 causal variants in the *PIAS4*, *PIK3C2G* and *DGAT1* genes colocalizes with an eQTL/sQTL SNP. However, as mentioned in the discussion, this may be due to a lack of power in the analyses, variation of gene expression at the time of sampling, or because the causal variant is not found in the breed used to perform the eQTL/sQTL analyses. This result does not invalidate their causal nature.

As requested, we have added a supplementary Figure 8 to depict the effects of the 3 causal variants on mRNA and protein structure, and how they impact mRNA stability and protein function. This has been mentioned in the results part of the main text, section *Identification of causal SDV*, L.406: “The effects of these three variants on mRNA transcripts and protein sequences with predicted functional impact are detailed in supplementary Fig. 8.”

Reviewer 3

Charles et al 2024 performed a comprehensive analysis of cattle splicing variants. Their work covered computational analyses and wet-lab experiments and identified a set of splice-disrupting variants which were followed by further annotations and/or validations. I found their work very novel and have set up an impressive example of combining computational analyses with wet-lab technologies to identify causal variants for the field. Therefore, I think their work has a very broad impact and is certainly worth being considered to be published in this journal. The manuscript is well written. I think the majority of the work is solid, although I do have several suggestions as follows that I hope the authors can consider to improve their manuscript.

It would be good to have a main table in the main text to show the details of those 38 SDVs, including rsID, chromosome, position, alleles, frequency and etc...

We have added an additional table in the main text describing the features of the 38 SDV, L.1246: “Table 3. The 38 SDV from the Var.GWAS dataset validated using Vex-seq”.

While I agree with the authors that some of those 38 SDVs are missed from the single-tissue sQTL analysis of modification of splicing rate in the cGTEx polite study (as in ref 6), I can find some overlaps between 38 SDVs and sQTL identified using intron splicing ratio from (ref 15, [https://www.cell.com/cell-genomics/fulltext/S2666-979X\(23\)00182-9](https://www.cell.com/cell-genomics/fulltext/S2666-979X(23)00182-9)) in the published data of https://figshare.unimelb.edu.au/articles/dataset/eQTL_and_sQTL_from_16_cattle_tissues_linear_mixe

d_model /19793047, such as blood, milk_cell and mammary. Also, I can find 31 of 38 SDVs are significant multi-tissue sQTL from [https://www.cell.com/cell-genomics/fulltext/S2666-979X\(23\)00182-9](https://www.cell.com/cell-genomics/fulltext/S2666-979X(23)00182-9) (in the folder of "MultiTissue.sQTL" from https://figshare.unimelb.edu.au/articles/dataset/eQTL_and_sQTL_from_16_cattle_tissues_linear_mixed_model_/19793047). I have listed those 31 identified multi-tissue sQTLs that are also SDVs below:

SNP smt.x2 smt.x2.p smtsq.x2 smtsq.x2.p
Chr6:83318905 0.542774226491673 0.461285265736643 254.899414725888 7.38656505464567e-26
Chr6:85199301 17.512042608103 2.85493617009921e-05 730.004962197001 7.13602689451379e-101
Chr6:85468855 58.714701153064 1.82274477306115e-14 75.1582244978063 1.47060375643707e-12
Chr6:85534747 58.714701153064 1.82274477306115e-14 75.1582244978063 1.47060375643707e-12
Chr2:1268402 14.6187464748533 0.000131598923466976 470.038959229041 2.85785527120729e-54
Chr3:15041705 8.64884453137295 0.00327268836985842 6894.23833144841 0
Chr3:15541059 59.4370566562152 1.26268856135547e-14 2544.33272202252 3.07856867767148e-283
Chr3:16482608 18.7621570297528 1.48076387164144e-05 1753.81138742094 8.1611383853056e-190
Chr4:76710798 34.0394380805022 5.4006264298877e-09 664.079208632038 3.08912535679002e-76
Chr5:119386807 53.396207660038 2.72622634767491e-13 252.066542795479 5.55445878422972e-31
Chr6:83318905 0.542774226491673 0.461285265736643 254.899414725888 7.38656505464567e-26
Chr6:85199301 17.512042608103 2.85493617009921e-05 730.004962197001 7.13602689451379e-101
Chr6:85468855 58.714701153064 1.82274477306115e-14 75.1582244978063 1.47060375643707e-12
Chr6:85534747 58.714701153064 1.82274477306115e-14 75.1582244978063 1.47060375643707e-12
Chr7:17999133 0.947173909503839 0.330439699046254 838.001202389464 3.04010994650235e-120
Chr11:103405870 7.03469669973857 0.00799454117299548 2379.889285441 3.80266085596523e-274
Chr14:1042682 10.0578068314694 0.00151703712538834 5341.55119964462 0
Chr14:1007601 123.556314484645 1.05353332706458e-28 3463.618665186 0
Chr14:1044777 10.1414458198163 0.00144972411139641 5566.22221715014 0
Chr14:446321 51.1213219397549 8.6829995072314e-13 2429.87420929955 2.08118331368934e-271
Chr14:1042664 51.5783641431177 6.87955619412734e-13 1980.74146962441 4.24717184531986e-229
Chr14:3033158 41.8861323003855 9.67463617490246e-11 865.627012964265 5.31514384539854e-119
Chr14:64187085 16.6426784880757 4.51241467144265e-05 1294.32523823418 8.46609551155823e-175
Chr14:1107890 7.77906365684522 0.00528551855590064 460.178075784977 1.70331285813676e-63
Chr15:77122984 1.59345127927204 0.206833734530018 2368.03709174989 6.66013944098851e-285
Chr15:77368805 29.6372180390019 5.20950148889296e-08 1894.01500558045 1.06358124061286e-217
Chr18:44773142 8.33302484978174 0.00389307813558396 840.050668075943 1.10578336319392e-100
Chr18:44900183 71.7569588618467 2.4340402958288e-17 1042.34995626398 1.10345701307671e-125
Chr19:40038954 4.25018523295684 0.0392460495858313 449.943292713611 2.2963129131019e-

Chr20:32759553 42.0323038787518 8.97779742344497e-11 1045.02260689244
 1.00551363633158e-150
 Chr20:33456383 157.691167773865 3.61511939411695e-36 1784.85595758747
 5.85143604352187e-268

I, therefore, encourage authors to check carefully with published data and to include these results in their paper and provide discussions. I think a key point is that the multi-tissue analysis is more powerful in identifying causal regulatory variants. Also, analysis of intron splicing ratio as a computational approach may be more powerful than traditionally used PSI. However, the multi-tissue sQTL is not clear which tissue these sQTLs are acting through, therefore, this is where the work from current authors can provide critical validations.

We thank reviewer 3 for this excellent suggestion, which we have incorporated in the revised manuscript.

First, we have specified in the material and method how we selected the eQTL/sQTL studies to identify eQTL/sQTL SNPs that colocalise with our 38 SDV, L.821:

“Comparison between SDV from Var.GWAS and eQTL/sQTL SNP

Studies reporting both eQTL and sQTL data in cattle were collected from PubMed. SNP shared between these studies and the 38 SDV from Var.GWAS were found using the rsID.”

Then, we have described the results, L.410:

“Colocalization between SDV from Var.GWAS and eQTL/sQTL SNP

We searched for shared SNP between the 38 SDV from Var.GWAS and published eQTL/sQTL data. Four studies did not allow the identification of common SNP^{74–77}. Two studies conducted on imputed WGS and involving large numbers of animals and samples showed positive results with respect to single-tissue eQTL/sQTL SNP^{7,25} and multi-tissue eQTL/sQTL SNP²⁵ (supplementary Table 6). SDV colocalize with 6 single-tissue eQTL SNP and 8 single-tissue sQTL SNP. Multi-tissue analysis yielded better results. Indeed, SDV also colocalized with 23 multi-tissue eQTL SNP and 23 multi-tissue sQTL SNP. Globally, 73.7% of SDV (27/38) colocalize with at least one single- or multi- e/sQTL SNP.”

Note that the detailed results of these analyses are available in the supplementary Table 6.

We also discussed these new results in the main text, L.525:

“Molecular QTL data from two studies showed that 27 out of the 38 SDV colocalize with cis-sQTL and cis-eQTL SNP in cattle^{7,25}. This argument supports the idea that these variants are true SDV. It should be noted that these two studies led to the identification of colocalized SDV as they were carried out on a large number of animals and samples, allowing the identification of many molecular QTL. Also, the multi-tissue analyses are more powerful and identify more QTL than single-tissue ones, which also explains why they colocalized with a greater number of SDV. Ten SDV did not colocalize with any molecular QTL, but this can be explained in various ways. The consequences of a SDV may be undetectable by sQTL or eQTL analyses if it is not expressed in the tissues tested or at the right timing. The power and completeness of these QTL analyses is also limited by the size and genetic background of the samples available.”

Of note, we identified 27 SDV which colocalize with eQTL/sQTL and not 31 as mentioned by Reviewer 3, because 4 variants are in duplicate in the list provided by reviewer 3 (Chr6:83318905, Chr6:85199301, Chr6:85468855, Chr6:85534747).

‘We discovered one new causal SDV (rs134725785) increasing DGAT1 expression and associated to modification of milk phenotypes.’ A previous study also reported that there is the splicing of a potential causal intron associated with milk traits (<https://www.biorxiv.org/content/10.1101/2022.07.13.499886v1.supplementary-material>, see attached table) which colocalises with rs134725785; this may be worthwhile reporting.

We have taken this information into consideration and mentioned it in the discussion with the suggested reference, L.551: “Finally, the rs134725785 polymorphism is located in the intron 2 of *DGAT1*, the splicing rate of which was associated with milk traits⁹⁷.”

Point-by-point response to the comments

We thank the reviewers for their useful comments and their positive evaluation of our work.

Minor comment's of Reviewer 1 are addressed below.

Note that only modifications relating to this second round of revision are highlighted in yellow. In addition, and as requested, all previous yellow marks have been removed from the supplementary information file.

Reviewer's comments

Reviewer #1 (Remarks to the Author):

I am pleased to see all my concerns have been addressed. I have just a few very minor comments in the attached version. Well done on a great manuscript.

L.62: replace "help identify the" by "the identification of"
This has been done. (L.61)

L.246: replace "allele" by "alleles"
This has been done. (L.245)

L.273: replace "to investigate" by "investigation of"
This has been done. (L.272)

L.281: Need year of publication in citation
The Sangermano study is cited above in the text (L.271), but the corresponding reference number [67] has been now also added L.281 for clarity: "from the study by Sangermano et al. (% of abnormal *ABCA4* transcripts associated with the ALT variant allele)⁶⁷"

L.405, 425, 537: insert "putative" before "causal variants"
We agree with Reviewer 1 that it is more cautious and relevant to use the term "putative" to describe the identified causal variants. The term "putative" has been added L.404, 424 and 536 as requested, and also L.133 and L.587.

L.538: "First, the rs134725785 SDV, which increased *DGAT1* expression, was associated with the modification of milk phenotypes". Provide a reference for this, Are you referring to this study or others or both?

The rs134725785 SNP is one of the 3 putative causal SDV identified in our study, but as pointed by Reviewer 1, this is not expressed clearly enough. We have reworded this passage to include all the information required for comprehension: "First, one of these three putative causal SDV was rs134725785. It increased *DGAT1* expression as determined by Vex-seq, and was associated with variation in milk phenotypes in our GWAS".